# CSPLoRA: Confidence-Guided Structural Planning for Low-Rank Adaptation

Huiming Ding [1]  Xiaochen Li [2]  Jianhui Ma [1]  Xu An [1]  Yihui Yang [1]  Zhenyu Tan [1]

## Abstract

Low-Rank Adaptation (LoRA) has become the de facto paradigm for parameter-efficient fine-tuning, with its effectiveness critically influenced by rank allocation across modules. However, existing approaches face a fundamental dilemma: uniform allocation ignores module heterogeneity, while adaptive methods introduce expensive training overhead or lack reusability across configurations. We propose CSPLoRA (Confidence-guided Structural Planning for LoRA), a decoupled framework that reweights probe samples by prediction uncertainty to obtain more discriminative module importance estimates. The key insight is that hard samples—those the model struggles with—provide more informative gradient signals for identifying critical modules than easy samples. For a fixed task-model pair, the resulting structural priors can be reused across compatible rank budgets and LoRA backends, supporting a practical "probe once, deploy everywhere" workflow. Experiments on GLUE, commonsense reasoning, and arithmetic tasks show that CSPLoRA improves over uniform LoRA on average (+1.25 points on LLaMA-2-7B commonsense reasoning) while maintaining comparable parameters, with the planned rank structure reusable across compatible LoRA variants.

## 1. Introduction

Large Language Models (LLMs) have demonstrated remarkable capabilities across diverse tasks (Brown et al., 2020; Chowdhery et al., 2023), but their massive scale poses significant challenges for adaptation. Full fine-tuning, which updates all parameters, is computationally prohibitive and often impractical for resource-constrained scenarios—requiring substantial memory, storage, and deployment costs. This has motivated the development of Parameter-Efficient Fine-Tuning (PEFT) methods (Houlsby et al., 2019; Li & Liang, 2021; Lester et al., 2021) that freeze pre-trained weights and introduce only a small number of trainable parameters for task-specific adaptation.

Among PEFT methods, Low-Rank Adaptation (LoRA) (Hu et al., 2022) has emerged as the de facto paradigm, inserting low-rank decomposition matrices into model layers while keeping original weights frozen. The success of LoRA has inspired numerous variants that improve its parameterization: DoRA (Liu et al., 2024) decouples magnitude and direction updates, PiSSA (Meng et al., 2024) leverages principal component analysis for initialization, LoRA+ (Hayou et al., 2024) uses differentiated learning rates, and VeRA (Kopiczko et al., 2024) reduces parameters via shared random matrices. However, these variants primarily focus on *how to parameterize* the low-rank updates, while a critical orthogonal factor remains underexplored: *how to allocate ranks across different modules*. Standard implementations assign uniform rank to all modules, ignoring the significant heterogeneity in module sensitivity to downstream tasks.

This uniform allocation is demonstrably suboptimal. Recent work has explored adaptive rank allocation strategies to address this issue. Training-coupled methods such as AdaLoRA (Zhang et al., 2023) and DyLoRA (Valipour et al., 2023) dynamically adjust ranks during training via importance-based pruning, achieving adaptive allocation but at the cost of expensive per-step overhead. Moreover, these approaches produce structures valid only for the current training run, requiring full retraining when exploring different rank budgets or deploying to different LoRA backends. More recently, decoupled approaches like GoRA (haonan he et al., 2025) plan structure before training using gradient-based importance estimation, reducing computational overhead. However, existing decoupled methods have primarily focused on *metric design*, while two complementary aspects remain unexplored.

**Confidence-guided probing for importance estimation.** Current gradient-based planning methods treat all probe samples equally when estimating module importance. However, samples exhibit natural heterogeneity in informativeness: samples the model already handles confidently con-

---

[1]University of Science and Technology of China, Hefei, Anhui, China [2]Information Engineering University, Zhengzhou, Henan, China. Correspondence to: Jianhui Ma <jianhui@ustc.edu.cn>.

*Proceedings of the 43rd International Conference on Machine Learning*, Seoul, South Korea. PMLR 306, 2026. Copyright 2026 by the author(s).

tribute redundant gradient information, while samples the model struggles with reveal which modules most urgently need adaptation. This asymmetry suggests that emphasizing hard samples during probing could yield more discriminative importance estimates. We complement existing metric-centric approaches with a *data-centric perspective*, exploring whether confidence-aware sample weighting can further improve structural planning.

**Reusable structural priors across configurations.** For a fixed task-model pair, we hypothesize that rank structure is largely governed by task characteristics and model architecture rather than a specific LoRA backend or rank budget. This suggests that a well-designed structural prior should transfer across compatible configurations. However, existing planning methods have primarily been evaluated within single configurations, leaving the question of cross-budget and cross-backend transferability underexplored. Systematic validation of such reusability would enable a truly decoupled "probe once, deploy everywhere" workflow within compatible settings, where a single probe amortizes its cost across multiple experimental configurations.

Based on these observations, we propose **CSPLoRA** (Confidence-guided Structural Planning for LoRA). Following the decoupled planning paradigm, CSPLoRA addresses both aspects through confidence-weighted probing and scale-invariant allocation. Our contributions are:

- We introduce a confidence-guided probing mechanism that reweights samples by prediction uncertainty, complementing existing metric-based approaches with a data-centric perspective.

- We design a scale-invariant allocation strategy that produces reusable structural priors, and systematically validate their transferability across compatible rank budgets and LoRA backends.

- Extensive experiments demonstrate broad improvements over uniform baselines across GLUE, commonsense reasoning, and arithmetic tasks.

**Conflict of Interest Disclosure.** The authors declare no financial conflicts of interest related to this work.

## 2. Related Work

**Rank Allocation Strategies for LoRA.** Low-Rank Adaptation (LoRA) (Hu et al., 2022) has become the dominant parameter-efficient fine-tuning paradigm, inspiring variants such as DoRA (Liu et al., 2024), PiSSA (Meng et al., 2024), LoRA+ (Hayou et al., 2024), QLoRA (Dettmers et al., 2023), and VeRA (Kopiczko et al., 2024). While these methods improve parameterization, they typically retain uniform rank allocation across modules. Recent work addresses this limitation through adaptive strategies. *Training-coupled* methods like AdaLoRA (Zhang et al., 2023) and DyLoRA (Valipour et al., 2023) dynamically adjust ranks during training via importance-based pruning or multi-rank training, achieving adaptive allocation at the cost of expensive overhead and limited reusability.

*Decoupled* approaches plan structure before training. GoRA (haonan he et al., 2025) estimates module importance via gradient information on probe data, fixing rank allocation for subsequent training. CSPLoRA complements this direction by exploring confidence-guided sample weighting during probing and validating reusable structural priors for compatible configurations under a fixed task-model pair.

**Importance Estimation in Neural Networks.** Identifying critical parameters is fundamental to model compression (Han et al., 2016) and efficient adaptation. Classical approaches like Optimal Brain Damage (LeCun et al., 1989) and Optimal Brain Surgeon (Hassibi et al., 1993) use second-order Hessian information, but their computational cost motivates first-order approximations. SNIP (Lee et al., 2019) and GraSP (Wang et al., 2020) leverage gradient-weight interactions (sensitivity-saliency) for pruning at initialization. In PEFT, existing methods (Zhang et al., 2023; haonan he et al., 2025) primarily focus on metric design (magnitude $|W|$, saliency $|W \cdot G|$, Fisher information). CSPLoRA complements this metric-centric view with a data-centric perspective: we show that confidence-weighted probing can produce more discriminative importance estimates for structural planning.

**Sample Weighting and Curriculum Learning.** Our confidence-guided probing relates to curriculum learning (Bengio et al., 2009), which organizes training samples from easy to hard. Recent work has explored sample difficulty estimation: focal loss (Lin et al., 2017) reweights samples by prediction confidence in object detection; uncertainty-based active learning selects informative samples for annotation. However, these techniques have primarily been applied to training dynamics rather than structural planning. CSPLoRA demonstrates that emphasizing hard samples during probing yields more discriminative module importance scores, complementing metric-based approaches with a data-centric perspective.

## 3. Method

This section presents the CSPLoRA framework. Our goal is to estimate module importance and derive effective rank allocation directly from pre-trained weights without training temporary adapters. As shown in Figure 1, the method consists of three stages: (1) Confidence-guided weighting that

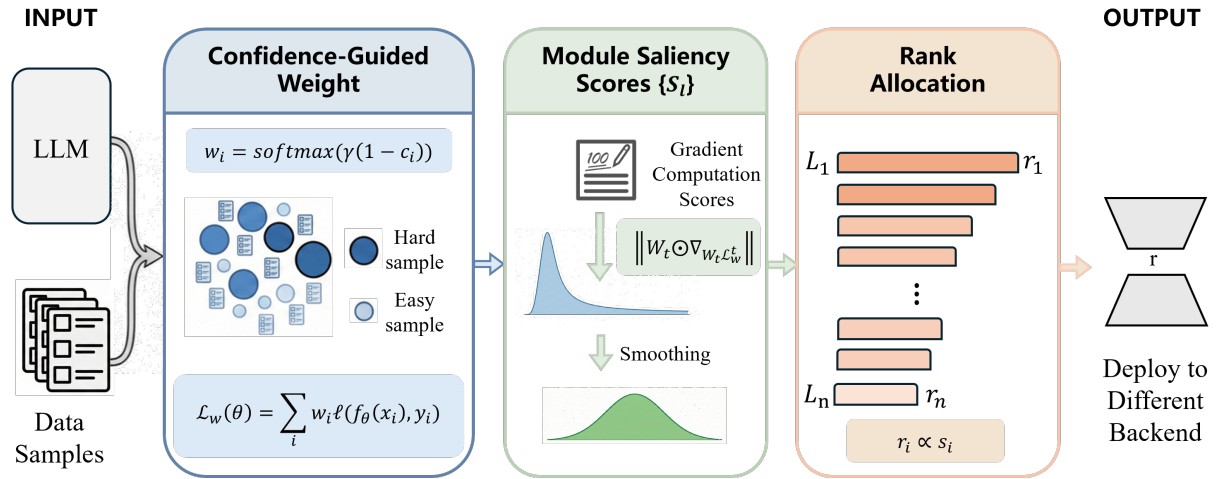

*Figure 1.* Overview of the CSPLoRA framework. Left: We construct a confidence-weighted risk $\mathcal{L}_w$ that assigns higher weights to hard samples (large circles) and lower weights to easy samples (small circles). Middle: Module saliency scores $\{S_\ell\}$ are computed via Taylor expansion on base weights, then smoothed to prevent concentration. Right: Ranks are allocated proportionally to smoothed scores, enabling deployment to different LoRA backends.

emphasizes hard samples; (2) Module saliency computation via Taylor expansion and smoothing; (3) Proportional rank allocation to target modules.

### 3.1. Problem Formalization

Let the pre-trained model parameters be $\theta_0$ and the downstream dataset be $\mathcal{D} = \{(x_i, y_i)\}_{i=1}^N$. We consider inserting LoRA adapters into $L$ target modules (e.g., attention projections and feed-forward layers) across the model. The standard fine-tuning objective is to find a set of optimal rank structures $\mathbf{r} = \{r_\ell\}_{\ell=1}^L$ under the constraint of total rank budget $R_{\text{tot}}$, such that the empirical risk after fine-tuning is minimized:

$$\min_{\mathbf{r}} \mathcal{L}_{std}(\theta) = \frac{1}{N} \sum_{i=1}^N \ell(f_\theta(x_i), y_i) \quad \text{s.t.} \sum_{\ell=1}^L r_\ell \leq R_{\text{tot}} \tag{1}$$

where $\ell(\cdot)$ is a task-specific loss function (e.g., cross-entropy). However, directly optimizing this objective usually requires expensive search costs. We transform this problem into two decoupled stages: first deriving budget-independent structural priors via importance estimation, then mapping these priors to specific rank allocations. This decoupling enables reusability across different experimental configurations.

### 3.2. Probing Stage: Confidence-Guided Importance Estimation

The core task of the probing stage is to quantify the potential contribution rate of each module to adapting to the downstream task. We achieve this through two components: confidence-weighted sampling to emphasize informative samples, and Taylor saliency to measure module-wise sensitivity.

**Confidence-Weighted Risk.** Standard empirical risk $\mathcal{L}_{std}$ treats all samples equally. However, "easy" samples that the model already predicts with high confidence contribute redundant gradient signals, whereas "hard" samples that the model struggles with reveal which modules urgently need adaptation. To focus computational resources on informative samples, we define confidence-weighted risk $\mathcal{L}_w$:

$$\mathcal{L}_w(\theta) = \sum_{i \in \mathcal{B}} w_i \cdot \ell(f_\theta(x_i), y_i) \tag{2}$$

where the sample weight $w_i$ is computed within each probe batch from the model's prediction confidence $c_i = p_\theta(y_i|x_i)$:

$$w_i = \frac{\exp(\gamma \cdot (1 - c_i))}{\sum_{j \in \mathcal{B}} \exp(\gamma \cdot (1 - c_j))}, \quad \gamma \geq 0 \tag{3}$$

The temperature parameter $\gamma$ controls the emphasis on hard samples: $\gamma = 0$ recovers uniform weighting, while larger $\gamma$ increasingly concentrates weight on low-confidence samples. A fixed $\gamma$ is simple, but a single global temperature may not match all probe batches because their difficulty spreads can differ. We therefore also consider an adaptive variant for diagnostic experiments. Let $d_i = 1 - c_i$ denote sample difficulty, $\bar{d}$ its batch mean, and $\sigma_\mathcal{B}$ its batch standard deviation. When $\sigma_\mathcal{B}$ is sufficiently large, we use

$$\gamma_{\text{eff}} = \min\left(\frac{\gamma_s}{\sigma_\mathcal{B}}, \gamma_{\max}\right), \quad w_i = \frac{\exp(\gamma_{\text{eff}}(d_i - \bar{d}))}{\sum_{j \in \mathcal{B}} \exp(\gamma_{\text{eff}}(d_j - \bar{d}))}, \tag{4}$$

and otherwise fall back to uniform weighting. This rule adapts the weighting strength to the observed difficulty dispersion without manually selecting a new global temperature. Appendix A provides a simplified variance perspective that motivates downweighting confidently solved samples.

**Taylor Saliency Metric.**    To evaluate the importance of layer $\ell$, we employ first-order Taylor expansion to approximate the loss change under parameter perturbation. Consider perturbing the weight matrix $W_\ell$ by $\Delta W_\ell$:

$$\mathcal{L}(W_\ell + \Delta W_\ell) - \mathcal{L}(W_\ell) \approx \sum_{i,j} \frac{\partial \mathcal{L}}{\partial W_\ell^{(i,j)}} \Delta W_\ell^{(i,j)} \quad (5)$$

When considering proportional perturbations $\Delta W_\ell^{(i,j)} = -\alpha W_\ell^{(i,j)}$ (i.e., shrinking weights toward zero), the first-order loss change becomes $\alpha \sum_{i,j} |W_\ell^{(i,j)} \cdot \nabla_{W_\ell}^{(i,j)} \mathcal{L}|$. This motivates our Taylor saliency metric, defined as the sum of element-wise products accumulated over $T$ probing batches:

$$\begin{aligned} S_\ell &= \frac{1}{T} \sum_{t=1}^{T} \sum_{(i,j)} \left| W_\ell^{(i,j)} \cdot \nabla_{W_\ell}^{(i,j)} \mathcal{L}_w^{(t)} \right| \\ &= \frac{1}{T} \sum_{t=1}^{T} \| W_\ell \odot \nabla_{W_\ell} \mathcal{L}_w^{(t)} \|_1 \end{aligned} \quad (6)$$

where $\|\cdot\|_1$ denotes the sum of absolute values of matrix elements. We provide analysis of this first-order approximation in Appendix A (Theorem A.3).

This aggregation produces a module-level saliency estimate before rank allocation. Importantly, the raw score is not directly used as the final rank: it is subsequently processed by score calibration, logarithmic smoothing, rank constraints, and budget projection. Thus, the final rank pattern reflects the complete planning pipeline rather than raw module size alone. We compare alternative importance metrics under the same pipeline in §4.4.2.

### 3.3. Structural Planning: Logarithmic Smoothing

Raw Taylor saliency scores $\{S_\ell\}$ typically exhibit heavy-tailed distributions, spanning multiple orders of magnitude across layers. Direct proportional allocation based on raw scores would lead to extreme "winner-takes-all" scenarios where a small fraction of high-scoring layers absorbs most of the rank budget, leaving other layers underparameterized.

Before smoothing, we apply a lightweight score calibration to keep the allocation from being dominated by raw scale differences across target-module families; for simplicity, we continue to denote the calibrated scores as $S_\ell$. To address heavy-tailed score distributions, we then apply a logarithmic smoothing transformation:

$$\tilde{s}_\ell = \log(1 + S_\ell) \quad (7)$$

This transformation has two key properties: (1) *monotonicity*—it preserves the relative ordering of layers, ensuring high-importance layers still receive larger ranks; (2) *tail compression*—the logarithm compresses heavy tails, preventing excessive concentration.

We analyze the concentration phenomenon in Appendix A (Proposition A.4), showing that when raw scores span multiple orders of magnitude, direct proportional allocation leads to extreme concentration. Logarithmic smoothing mitigates this issue while maintaining discriminative power.

### 3.4. Rank Allocation and Reuse

Given the smoothed importance scores $\{\tilde{s}_\ell\}$, we map them to integer ranks via a "baseline + proportional boost" strategy:

$$r_\ell(R_{\text{tot}}) = r_{\min} + \left\lfloor (R_{\text{tot}} - L \cdot r_{\min}) \cdot \frac{\tilde{s}_\ell}{\sum_{k=1}^{L} \tilde{s}_k} \right\rfloor \quad (8)$$

where $r_{\min}$ is a minimum rank ensuring all layers maintain basic expressivity, and $R_{\text{tot}}$ is the total rank budget. This allocation ensures that each layer receives at least $r_{\min}$ rank, with the remaining budget distributed proportionally to importance scores.

We also introduce an optional coverage parameter $\rho \in (0, 1]$ that allocates the excess budget only to the top-$\rho$ fraction of layers by importance, with remaining layers receiving $r_{\min}$. Ablation experiments (Appendix B.1) show that $\rho = 1.0$ (full coverage) performs well, so we use this as default.

We prove in Appendix A (Proposition A.5) that under concave utility assumptions (diminishing marginal returns), this proportional allocation is optimal in the sense of maximizing total adaptation utility.

**Scale Invariance and Reusability.**    A key property of this allocation is *scale invariance*: for any two layers $i, j$, the ratio of their rank increments is:

$$\frac{r_i - r_{\min}}{r_j - r_{\min}} = \frac{\tilde{s}_i}{\tilde{s}_j} \quad (9)$$

which is independent of the total budget $R_{\text{tot}}$. For a fixed task-model pair, this means the structural prior (relative importance ordering and ratios) can be computed once and reused across different rank budgets.

Furthermore, because CSPLoRA only fixes the rank pattern and does not change the adapter parameterization itself, the same structural prior can be applied to compatible LoRA variants. This supports a practical "probe once, deploy everywhere" workflow across compatible configurations.

The reusable object is the calibrated module-importance profile, not a set of trained adapter weights. Therefore, reuse

*Table 1.* Fine-tuning performance of LLaMA-2-7B and LLaMA-3.1-8B on 8 commonsense reasoning tasks. Results report mean ± std over 3 random seeds. **Bold** denotes the better result in each backend/CSPLoRA pair.

| Model | Backend Method | BoolQ | HeSw | PIQA | ARC-C | ARC-E | OBQA | SIQA | WiGr | Avg |
|---|---|---|---|---|---|---|---|---|---|---|
| | Full FT | 72.18±0.39 | 89.56±1.08 | 83.06±0.36 | 71.47±1.32 | 84.16±0.07 | 82.67±0.95 | 79.72±0.74 | 83.66±0.65 | 80.81 |
| | AdaLoRA | 65.29±0.52 | 63.37±6.68 | 78.31±0.55 | 60.69±0.77 | 79.53±0.75 | 67.13±0.42 | 74.72±0.22 | 72.30±0.28 | 70.17 |
| | GoRA | 70.08±0.18 | 85.19±2.20 | 80.54±0.49 | 65.19±0.45 | 82.98±0.41 | 76.80±0.92 | 77.67±0.23 | 79.58±0.32 | 77.25 |
| | LoRA | 67.43±0.70 | 75.12±4.99 | 79.13±0.58 | 62.60±0.26 | 80.82±0.47 | 71.73±0.42 | 76.24±0.77 | 74.82±0.57 | 73.49 |
| | CSP-LoRA | 67.86±0.61 | 78.44±1.93 | 79.62±0.11 | 63.59±0.69 | 81.55±0.24 | 73.80±1.06 | 76.80±0.49 | 76.30±0.18 | **74.74** |
| **LLaMA-2-7B** | LoRA+ | 69.33±1.17 | 83.56±4.62 | 81.66±0.26 | 66.35±0.53 | 82.94±0.07 | 79.77±1.05 | 79.59±0.68 | 81.28±0.66 | 78.06 |
| | CSP-LoRA+ | 70.50±0.66 | 86.02±1.61 | 81.96±0.17 | 67.46±0.31 | 83.46±0.42 | 77.57±0.56 | 78.72±0.08 | 79.91±0.47 | **78.20** |
| | DoRA | 67.79±0.85 | 74.58±5.19 | 79.49±0.39 | 62.57±0.13 | 80.88±0.17 | 71.40±0.72 | 76.03±0.42 | 75.14±0.48 | 73.48 |
| | CSP-DoRA | 68.31±0.51 | 80.24±1.91 | 79.67±0.31 | 63.82±0.39 | 81.38±0.56 | 72.87±0.42 | 76.63±0.26 | 76.40±0.16 | **74.91** |
| | PiSSA | 69.69±0.27 | 83.06±2.10 | 80.13±0.60 | 64.33±0.35 | 82.75±0.74 | 75.60±0.69 | 77.18±0.66 | 77.27±0.28 | 76.25 |
| | CSP-PiSSA | 69.68 ± 0.55 | 83.99 ± 1.04 | 81.39 ± 0.09 | 64.85 ± 0.52 | 82.93 ± 0.28 | 77.00 ± 1.11 | 77.64 ± 0.11 | 78.30 ± 0.44 | **76.97** |
| | Full FT | 74.21±0.12 | 95.79±0.07 | 88.88±0.61 | 82.51±0.56 | 91.72±0.28 | 87.47±0.42 | 80.62±0.13 | 87.53±0.29 | 86.09 |
| | AdaLoRA | 69.36±0.35 | 90.84±0.59 | 84.80±0.65 | 77.25±1.55 | 90.12±0.27 | 79.27±0.12 | 75.90±0.57 | 79.56±0.83 | 80.89 |
| | GoRA | 74.03±0.77 | 95.40±0.15 | 88.47±0.80 | 80.57±0.50 | 90.80±0.78 | 85.87±0.50 | 80.11±0.43 | 86.58±0.08 | 85.23 |
| | LoRA | 72.53±1.26 | 94.35±0.05 | 88.03±0.14 | 79.80±0.55 | 90.64±0.17 | 84.47±1.75 | 79.24±0.39 | 85.24±0.65 | 84.29 |
| | CSP-LoRA | 72.90 ± 0.66 | 94.82 ± 0.17 | 88.05 ± 0.41 | 79.44 ± 0.09 | 90.87 ± 0.36 | 86.07 ± 0.43 | 79.79 ± 0.44 | 85.85 ± 0.58 | **84.72** |
| **LLaMA-3.1-8B** | LoRA+ | 73.10±0.42 | 94.87±0.15 | 88.21±0.19 | 79.83±0.13 | 90.94±0.38 | 85.00±0.53 | 79.68±0.09 | 85.87±0.72 | **84.69** |
| | CSP-LoRA+ | 72.84±0.66 | 94.84±0.19 | 87.56±0.41 | 78.54±0.36 | 90.02±0.35 | 84.70±0.62 | 80.16±0.08 | 86.89±0.51 | 84.44 |
| | DoRA | 72.60±0.70 | 94.55±0.09 | 87.72±0.37 | 79.89±0.66 | 90.59±0.29 | 84.20±1.25 | 79.27±0.18 | 85.98±0.40 | 84.35 |
| | CSP-DoRA | 72.83±0.73 | 94.72±0.20 | 88.29±0.59 | 79.58±0.34 | 90.64±0.26 | 84.87±0.76 | 79.36±0.26 | 85.63±0.82 | **84.49** |
| | PiSSA | 70.18±2.87 | 93.07±1.73 | 85.91±1.23 | 76.31±3.22 | 88.36±2.17 | 82.87±3.35 | 78.51±1.64 | 84.24±2.24 | 82.43 |
| | CSP-PiSSA | 72.38±1.58 | 94.92±0.23 | 87.94±0.19 | 79.55±0.82 | 90.18±0.18 | 84.33±0.81 | 79.60±0.23 | 86.27±0.48 | **84.40** |

is appropriate when the downstream task, base model, and target-module set are fixed, while the user varies compatible rank budgets or LoRA backends. If the task distribution, model architecture, or target-module definition changes, CSPLoRA performs a new probe to avoid transferring a mismatched structural prior. This distinction keeps the planner decoupled from final adapter optimization while making the amortized probe cost meaningful in realistic hyperparameter and backend exploration.

## 4. Experiments

We conducted comprehensive comparisons of CSPLoRA against baseline methods on Natural Language Understanding (GLUE), commonsense reasoning, and arithmetic reasoning tasks. Our goal is to demonstrate that CSPLoRA is a plug-and-play structural planner that provides effective rank allocation for compatible LoRA backends without the training overhead of dynamic methods.

### 4.1. Experimental Setup

**Baselines.** We compare CSPLoRA against two categories of methods: (1) **Uniform rank baselines**: LoRA (Hu et al., 2022), LoRA+ (Hayou et al., 2024), DoRA (Liu et al., 2024), and PiSSA (Meng et al., 2024). These methods allocate the same rank $r$ to all modules, serving as strong baselines to isolate the effect of structural planning under comparable average-rank budgets. (2) **Adaptive rank allocation**: AdaLoRA (Zhang et al., 2023) and GoRA (haonan he et al., 2025). AdaLoRA dynamically adjusts ranks during train-

ing via SVD pruning, while GoRA plans structure before training using gradient information. These represent strong non-uniform allocation baselines. We also report full fine-tuning as a reference upper baseline where applicable.

**Notation and comparison protocol.** CSPLoRA is a rank planner rather than a new adapter parameterization. We therefore denote applying the CSPLoRA planner to a backend as CSP-LoRA, CSP-PiSSA, CSP-DoRA, or CSP-LoRA+. In each paired comparison, the backend, optimizer, training schedule, and target modules are kept the same; only the rank pattern changes from uniform allocation to the CSPLoRA-planned structure. This protocol isolates the effect of structural planning while allowing the same planned structure to be deployed with compatible LoRA backends.

**Implementation.** **Models**: We evaluate on LLaMA-2-7B (Touvron et al., 2023), LLaMA-3.1-8B (Grattafiori et al., 2024), and RoBERTa-large (Liu et al., 2019) (a robustly optimized BERT (Devlin et al., 2019) variant). **Training Protocol**: Following haonan he et al. (2025), we report mean and standard deviation over 3 random seeds for the main experiments. **Probing**: We employ grouped low-memory mode (*full-lowmem*) to compute layer-wise gradients sequentially, reducing peak memory usage.

**Tasks**: We evaluate on three domains: (i) *Commonsense reasoning*: train on Commonsense170K (Hu et al., 2023), evaluate on 8 benchmarks (BoolQ (Clark et al., 2019), HellaSwag (Zellers et al., 2019), PIQA (Bisk et al., 2020), ARC-Challenge/Easy (Clark et al., 2018), OpenbookQA (Mi-

*Table 2.* Results of RoBERTa-large on GLUE dev set. Results report mean ± std over 3 random seeds. **Bold** denotes the better result in each backend/CSPLoRA pair.

| Backend Method | MNLI | QQP | QNLI | SST-2 | CoLA | MRPC | RTE | STS-B | Avg |
|---|---|---|---|---|---|---|---|---|---|
| Full FT | 90.31±0.22 | 89.37±0.14 | 94.62±0.17 | 96.41±0.18 | 67.41±1.27 | 93.25±0.55 | 85.68±0.55 | 92.08±0.21 | 88.64 |
| AdaLoRA | 90.75±0.10 | 86.75±0.11 | 94.48±0.15 | 95.72±0.18 | 62.50±0.63 | 90.75±1.29 | 70.04±1.57 | 91.40±0.21 | 85.3 |
| GoRA | 90.53±0.04 | 86.50±0.09 | 94.28±0.25 | 95.64±0.11 | 63.07±0.65 | 91.95±0.58 | 81.41±1.28 | 89.22±2.51 | 86.57 |
| LoRA | 90.32±0.08 | 85.92±0.18 | 93.64±0.04 | 95.41±0.11 | 63.25±1.23 | 91.21±0.22 | 71.48±4.16 | 91.16±0.17 | 85.3 |
| CSP-LoRA | 90.44±0.06 | 86.03±0.08 | 93.90±0.21 | 95.64±0.20 | 64.17±2.05 | 91.05±0.76 | 77.26±0.51 | 91.25±0.24 | **86.21** |
| LoRA+ | 90.56±0.05 | 87.35±0.13 | 94.57±0.13 | 95.76±0.11 | 66.15±0.18 | 92.53±0.35 | 82.07±1.16 | 91.40±0.63 | 87.55 |
| CSP-LoRA+ | 90.61±0.06 | 87.55±0.10 | 94.68±0.21 | 95.87±0.49 | 67.70±1.02 | 93.09±0.69 | 84.00±1.78 | 92.10±0.16 | **88.2** |
| DoRA | 90.78±0.22 | 87.61±0.17 | 94.75±0.12 | 96.14±0.29 | 69.29±0.93 | 93.25±0.90 | 84.96±1.46 | 92.19±0.04 | 88.62 |
| CSP-DoRA | 90.67±0.21 | 87.93±0.04 | 94.86±0.05 | 96.02±0.48 | 69.20±0.69 | 93.00±0.96 | 85.56±0.96 | 92.02±0.32 | **88.65** |
| PiSSA | 90.36±0.05 | 86.30±0.10 | 93.69±0.09 | 95.80±0.18 | 61.17±0.96 | 91.31±0.46 | 67.69±0.68 | 89.51±0.65 | 84.48 |
| CSP-PiSSA | 90.53±0.19 | 86.51±0.05 | 94.11±0.08 | 95.87±0.20 | 60.86±1.11 | 91.51±0.40 | 75.45±0.51 | 89.98±0.27 | **85.6** |

haylov et al., 2018), SIQA (Sap et al., 2019), Wino-Grande (Sakaguchi et al., 2021)); (ii) *Arithmetic reasoning*: train on MetaMathQA-50K (Yu et al., 2024), evaluate on GSM8K (Cobbe et al., 2021) and MATH (Hendrycks et al., 2021); (iii) *Natural language understanding*: fine-tune on GLUE (Wang et al., 2018) dev sets.

**Infrastructure**: Main experiments run on NVIDIA A100/A800 GPUs using PyTorch (Paszke et al., 2019) and HuggingFace Transformers (Wolf et al., 2020). Additional diagnostics run on an RTX 5090 environment are reported separately; we only compare methods within the same diagnostic table instead of merging them with the main results. Full hyperparameters appear in Appendix D.

### 4.2. Main Results

#### 4.2.1. COMMONSENSE REASONING

We evaluate on LLaMA-2-7B and LLaMA-3.1-8B across 8 commonsense reasoning benchmarks. As shown in Table 1, CSPLoRA generally improves over uniform baselines across multiple backends.

On LLaMA-2-7B, all CSPLoRA-planned backends improve over their uniform counterparts: CSP-LoRA improves the average from 73.49 to 74.74, CSP-DoRA from 73.48 to 74.91, and CSP-PiSSA from 76.25 to 76.97. On LLaMA-3.1-8B, CSPLoRA also improves LoRA, DoRA, and PiSSA, while LoRA+ is already a strong backend and shows a slight decrease. These results indicate that rank planning is most beneficial when the uniform backend leaves clear allocation inefficiency.

#### 4.2.2. GLUE BENCHMARK

Table 2 presents results on RoBERTa-large across 8 GLUE tasks. CSPLoRA improves the average performance across all backends, with particularly notable gains on challenging tasks.

CSP-LoRA improves the average score from 85.30 to 86.21 (+0.91), with the largest gain on RTE (+5.78), a task where the uniform LoRA baseline has high variance. When combined with stronger backends, CSPLoRA further narrows the gap to full fine-tuning: CSP-DoRA reaches 88.65, comparable to full fine-tuning at 88.64. These results suggest that structural rank planning is complementary to backend-level parameterization improvements.

*Table 3.* Performance on arithmetic reasoning tasks (GSM8K and MATH). Results report mean ± std over 3 random seeds. **Bold** denotes the better result in each backend/CSPLoRA pair.

| Model | Backend Method | GSM8K | MATH |
|---|---|---|---|
| **LLaMA-2-7B** | Full FT | 50.21±0.58 | 8.17±0.24 |
| | AdaLoRA | 34.11±0.60 | 5.07±0.14 |
| | GoRA | 42.61±0.87 | 6.67±0.13 |
| | LoRA | 39.47±1.22 | 5.87±0.42 |
| | CSP-LoRA | **40.91±0.66** | **6.13±0.44** |
| | LoRA+ | 48.30±0.17 | 7.28±0.52 |
| | CSP-LoRA+ | **48.72±0.14** | **7.61±0.17** |
| | PiSSA | 41.42±0.23 | 6.57±0.27 |
| | CSP-PiSSA | **43.77±1.04** | **6.85±0.04** |
| **LLaMA-3.1-8B** | Full FT | 75.26±0.38 | 24.69±0.55 |
| | AdaLoRA | 64.54±0.37 | 20.53±0.38 |
| | GoRA | 72.91±0.84 | 24.74±0.25 |
| | LoRA | 71.64±0.75 | 24.36±0.47 |
| | CSP-LoRA | **72.33±0.20** | **24.38±0.30** |
| | LoRA+ | 71.94±0.53 | **24.33±0.38** |
| | CSP-LoRA+ | **72.66±0.20** | 24.26±0.17 |
| | PiSSA | 70.81±0.92 | 22.28±0.41 |
| | CSP-PiSSA | **71.05±0.98** | **22.86±0.54** |

#### 4.2.3. ARITHMETIC REASONING

Table 3 reports results on GSM8K and MATH, two benchmarks requiring multi-step mathematical reasoning. CSPLoRA generally yields gains across model scales and backends.

On LLaMA-2-7B, CSPLoRA improves all three evaluated LoRA-style backends on both GSM8K and MATH. The gains are largest for PiSSA, where CSP-PiSSA improves GSM8K from 41.42 to 43.77 and MATH from 6.57 to 6.85. This indicates that structural rank planning can complement stronger initialization-based backends rather than only improving vanilla LoRA.

On LLaMA-3.1-8B, the gains become smaller but remain visible on GSM8K: CSPLoRA improves LoRA, LoRA+, and PiSSA by +0.69, +0.72, and +0.24 points, respectively. On MATH, improvements are backend-dependent, with gains for LoRA and PiSSA but a slight decrease for LoRA+. These results suggest that the benefit of rank planning depends on how much allocation inefficiency remains in the backend and task, while still transferring across two LLaMA generations.

## 4.3. Analysis

We analyze the structural patterns discovered by CSPLoRA's probing mechanism and examine its convergence behavior.

### 4.3.1. SALIENCY SCORE PATTERNS

Figure 2 visualizes the Taylor saliency scores across layers and module types for LLaMA-2-7B. Two patterns emerge:

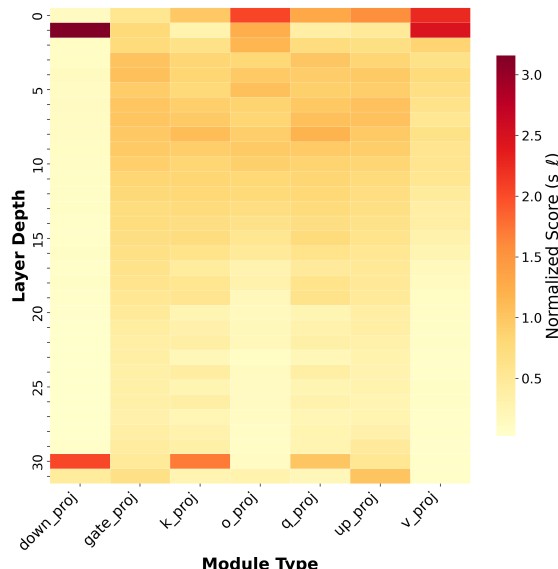

*Figure 2.* Taylor saliency heatmap for LLaMA-2-7B on commonsense reasoning. Y-axis: layer depth (0–31); X-axis: module types. Darker colors indicate higher importance.

**Depth-wise heterogeneity.** Boundary layers receive relatively higher scores than many middle layers, especially for down$_{proj}$, o$_{proj}$, and v$_{proj}$. This suggests that the probe captures depth-dependent sensitivity: modules near the input and output sides may play different roles from those in the middle of the network during task adaptation.

**Module-wise heterogeneity.** The score distribution also varies across module types. Attention output/value projections and FFN down projections tend to receive higher scores, while key projections are less emphasized in this diagnostic. These differences indicate that module sensitivity is not well described by a uniform rank assumption.

Overall, the heatmap supports the motivation for structured rank planning. CSPLoRA uses these score patterns as planning signals to derive a constrained rank allocation, allowing the final rank structure to reflect both layer-wise and module-wise heterogeneity while remaining under the target budget.

### 4.3.2. PROBE CONVERGENCE

Figure 3 analyzes how quickly importance scores stabilize during probing. We measure relative change as $\|\mathbf{s}^{(t)} - \mathbf{s}^{(t-10)}\|_2 / \|\mathbf{s}^{(t-10)}\|_2$ across probe steps. The curve shows rapid initial convergence: relative change drops from 17% at step 20 to below the 2% threshold by step 40. After step 40, fluctuations remain under 5%, indicating that the score profile becomes relatively stable in this setting. This supports using 50–100 probe steps as a practical default: the planner can obtain stable structural estimates without extensive probing.

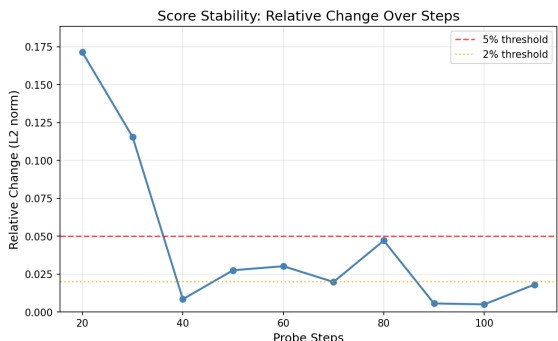

*Figure 3.* Probe convergence on LLaMA-2-7B commonsense tasks. Relative change (measured every 10 steps) drops below 2% by step 40 and stabilizes thereafter, validating our default of 50–100 probe steps.

## 4.4. Ablation Studies

We conduct ablations to isolate the core design choices in CSPLoRA. Unless otherwise noted, experiments use LLaMA-2-7B on a 5-task commonsense subset (BoolQ, HellaSwag, PIQA, ARC-C, WinoGrande), reporting average accuracy. This subsection focuses on pipeline components and importance metrics, while parameter sensitivity and reuse/efficiency analyses are separated below.

### 4.4.1. EFFECT OF STRUCTURAL COMPONENTS

Table 4 evaluates the necessity of score calibration and smoothing. Removing either component degrades performance, with smoothing being particularly critical (73.51 → 73.08). Without smoothing, raw Taylor scores exhibit extreme disparity (spanning orders of magnitude), leading to pathological "winner-takes-all" allocation where most budget concentrates in a few top-scoring layers. Logarithmic smoothing compresses this heavy tail while preserving relative ordering, enabling balanced yet discriminative rank allocation.

*Table 4.* Ablation of structural components.

| Configuration | Avg Accuracy |
|---|---|
| CSPLoRA (Full) | **73.51** |
| w/o Calibration | 73.18 |
| w/o Smoothing | 73.08 |
| w/o Both (Raw Taylor) | 72.98 |

### 4.4.2. COMPARISON OF IMPORTANCE METRICS

Table 5 compares three importance metrics under the same allocation pipeline: (i) Fisher information ($\|G\|_F^2$), (ii) mean Taylor saliency (used in GoRA), and (iii) sum Taylor saliency. Sum Taylor gives the best average in this ablation, although the gap to mean Taylor is small.

These results should be interpreted at the pipeline level rather than as a statement about raw score magnitude alone. The final rank pattern is obtained after score calibration, smoothing, minimum-rank constraints, and budget projection, so raw module saliency is not directly equivalent to trainable parameter allocation. We provide an additional parameter audit in Appendix C.

*Table 5.* Comparison of importance metrics under the same allocation pipeline.

| Metric | Avg Accuracy |
|---|---|
| Fisher ($\|G\|_F^2$) | 73.37 |
| Mean Taylor | 73.47 |
| Sum Taylor (Ours) | **73.51** |

### 4.5. Parameter Sensitivity

We next examine hyperparameters that control the behavior of the rank planner. These experiments are separated from the component ablations because they do not remove parts of CSPLoRA; instead, they study how planning choices affect the final allocation and performance.

### 4.5.1. CONFIDENCE TEMPERATURE AND ADAPTIVE WEIGHTING

Table 6 studies the effect of the fixed confidence temperature $\gamma$ in the original 5-task setting. Uniform weighting ($\gamma = 0$) is weaker than confidence-aware variants, while a moderate fixed temperature performs best among the tested fixed choices. These results support the use of confidence weighting, but they also show that the temperature has a measurable effect on planning performance.

Because a single fixed temperature may not fit every local difficulty distribution, we additionally evaluate the adaptive rule from Section 3.2 in Table 7. This diagnostic is run on RTX 5090 and should be interpreted only within the table: all rows use the same environment and evaluation subset, but the numbers are not merged with the A100/A800 main results. The adaptive variant sets the effective temperature from batch-level difficulty dispersion, reducing reliance on manually selecting a single global $\gamma$, and gives the strongest within-table average.

*Table 6.* Effect of confidence temperature $\gamma$.

| $\gamma$ | Avg Accuracy |
|---|---|
| 0.0 (Uniform) | 73.33 |
| 1.0 | 73.43 |
| 2.0 (Default) | **73.51** |
| 4.0 | 73.48 |

*Table 7.* Adaptive confidence-weighting diagnostic.

| Setting | 5-task Avg |
|---|---|
| LoRA | 73.59 |
| 0.0 (Uniform) | 73.95 |
| 2.0 (Fixed) | 74.04 |
| Adaptive | **74.40** |

### 4.6. Reusability and Efficiency

We finally analyze whether the structural prior can be reused across compatible configurations and whether the decoupled probe introduces meaningful overhead.

### 4.6.1. CROSS-BUDGET AND CROSS-BACKEND REUSABILITY

A key advantage of CSPLoRA is that importance scores reflect task-specific module sensitivity, which is determined by the task and model architecture. This enables two forms of reusability:

Cross-budget reuse. Table 8 shows that a single probe generalizes across rank budgets ($r \in \{4, 8, 16\}$). CSPLoRA outperforms uniform allocation across the tested budgets, with the largest gain at $r = 8$ (+1.69). This scale invariance

arises from our proportional allocation: relative importance ratios are budget-independent.

Cross-backend reuse. All main experiments (Tables 1–3) apply the same probed structural prior to LoRA, LoRA+, DoRA, and PiSSA within the corresponding task-model setting. Consistent improvements across compatible backends indicate that rank planning captures information complementary to the specific LoRA parameterization.

Table 8. Cross-budget reuse: single probe, multiple budgets.

| Avg Rank | Uniform | CSPLoRA |
|----------|---------|---------|
| $r = 4$ | 71.05 | **71.48** (+0.43) |
| $r = 8$ | 71.82 | **73.51** (+1.69) |
| $r = 16$ | 75.01 | **75.15** (+0.14) |

#### 4.6.2. COMPUTATIONAL EFFICIENCY

We analyze parameter efficiency and computational overhead relative to adaptive baselines.

Parameter efficiency. Table 9 reports trainable parameters at $r = 8$. CSPLoRA uses over 30% fewer parameters than AdaLoRA (which allocates $1.5\times$ parameter space for masking) while matching uniform LoRA. For LLaMA-2-7B, CSPLoRA requires 19.77M parameters vs. AdaLoRA's 29.99M. Unlike AdaLoRA's dynamic masking, CSPLoRA fixes rank allocation before training, avoiding masking overhead.

Table 9. Parameter counts at rank budget $r = 8$.

| Method | RoBERTa | LLaMA-2 | LLaMA-3.1 |
|--------|---------|---------|-----------|
| LoRA | 3.21M | 19.99M | 20.97M |
| AdaLoRA | 4.30M | 29.99M | 31.46M |
| GoRA | 3.16M | 20.18M | **20.02M** |
| CSPLoRA | **3.01M** | **19.77M** | 20.76M |

Computational overhead. Table 10 decomposes time into probe and training phases. Probing takes 11–67s (<1% of training time), without dynamic planner-specific operations during final training—after the rank pattern is fixed, CSPLoRA follows the same training procedure as the chosen LoRA backend and introduces no dynamic rank-update or SVD overhead. Crucially, probe cost amortizes across compatible budgets and backends: one 67s probe enables experimentation with multiple $r$ values and LoRA variants. In contrast, AdaLoRA incurs continuous SVD overhead during training.

Table 10. Time breakdown on A100 GPU.

| Phase | RoBERTa | LLaMA-2 | LLaMA-3.1 |
|-------|---------|---------|-----------|
| Probe | 11s | 62s | 67s |
| Train | 230s | 4.54h | 9.19h |

## 5. Conclusion

We presented CSPLoRA, a confidence-guided structural planning framework for LoRA that estimates module importance *before* training. The key idea is that hard samples—those the model struggles with—provide more informative gradient signals for identifying critical modules than easy samples. Combined with scale-invariant allocation, CSPLoRA supports a "probe once, deploy everywhere" workflow by reusing structural priors across compatible rank budgets and LoRA backends for a fixed task-model pair. Experiments on GLUE, commonsense reasoning, and arithmetic tasks show broad average improvements over uniform baselines across multiple LoRA variants. CSPLoRA complements existing work on LoRA parameterization by providing an orthogonal perspective on structural planning. Code is available at https://github.com/Ming-to/CSPLoRA.git.

## Limitations

- As a structural planning method, CSPLoRA is inherently limited by the expressiveness of the underlying LoRA backend. While it often improves over uniform allocation, it cannot fundamentally overcome the capacity constraints of low-rank parameterization.

- Current evaluation focuses on text-only tasks; extending to vision-language models where rank allocation may need to balance visual encoders and language components remains unexplored.

- Appendix C includes additional T5-base diagnostics, but broader validation on larger encoder-decoder models and instruction-tuning settings remains valuable.

## Impact Statement

This work aims to make task adaptation of large language models more parameter-efficient by improving how low-rank adaptation capacity is allocated across modules. By reducing unnecessary trainable parameters and enabling structural plans to be reused across compatible configurations, CSPLoRA may lower the computational cost of adapting large models and make experimentation more accessible to researchers with limited hardware resources. The method does not introduce new model capabilities by itself, but its societal impact depends on the base models and downstream tasks to which it is applied; sensitive applications should still be evaluated for bias, safety, and misuse risks.

## Acknowledgements

This work was supported in part by the National Natural Science Foundation of China (No. U23A20319, 62441239).

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

# A. Theoretical Analysis

This appendix provides supporting analysis for CSPLoRA's design choices. The goal is to motivate the design rather than provide a general guarantee for all data distributions. We analyze: (1) how confidence weighting rebalances sample contributions under a simplified estimator, (2) why Taylor saliency $\sum |W \odot G|$ quantifies layer sensitivity to proportional perturbations, (3) how logarithmic smoothing prevents pathological rank concentration, and (4) the natural properties satisfied by proportional rank allocation.

**Confidence-Weighted Importance Estimation.** We formalize how confidence weighting downweights confidently solved samples under a simplified gradient-estimation model.

**Proposition A.1** (Adaptive sample reweighting). *Consider a batch $\mathcal{B}$ with samples having confidences $\{c_i\}_{i \in \mathcal{B}}$ where $c_i = p_\theta(y_i|x_i) \in (0,1)$. Under confidence weighting with temperature $\gamma > 0$, the weight assigned to sample $i$ is:*

$$w_i = \frac{\exp(\gamma(1-c_i))}{\sum_{j \in \mathcal{B}} \exp(\gamma(1-c_j))} \tag{10}$$

*For a sample with confidence $c_i$ compared to the batch mean confidence $\bar{c} = \frac{1}{|\mathcal{B}|} \sum_j c_j$, by Jensen's inequality applied to the convex function $\exp(\cdot)$:*

$$w_i \leq \frac{1}{|\mathcal{B}|} \exp(\gamma(\bar{c} - c_i)) \tag{11}$$

*Proof.* The partition function satisfies:

$$Z = \sum_{j \in \mathcal{B}} \exp(\gamma(1-c_j)) \geq |\mathcal{B}| \cdot \exp(\gamma(1-\bar{c})) \tag{12}$$

with equality when all confidences are identical. Therefore:

$$w_i = \frac{\exp(\gamma(1-c_i))}{Z} \leq \frac{\exp(\gamma(1-c_i))}{|\mathcal{B}| \exp(\gamma(1-\bar{c}))} = \frac{1}{|\mathcal{B}|} \exp(\gamma(\bar{c} - c_i)) \tag{13}$$

For $c_i > \bar{c}$: $\bar{c} - c_i < 0$, so $w_i < 1/|\mathcal{B}|$. For $c_i < \bar{c}$: $\bar{c} - c_i > 0$, so $w_i > 1/|\mathcal{B}|$. $\qquad\square$

**Theorem A.2** (Easy-sample downweighting under a simplified estimator). *Consider estimating layer-wise importance using weighted gradients $\hat{S}_\ell = \sum_{i \in \mathcal{B}} w_i g_{\ell,i}$ where $g_{\ell,i} = \|W_\ell \odot G_{\ell,i}\|_1$ is the Taylor saliency for sample $i$ at layer $\ell$. Under the assumption that samples $\{g_{\ell,i}\}$ are i.i.d. with mean $\mu = \mathbb{E}[g_\ell]$ and variance $\sigma^2 = \text{Var}[g_\ell]$, and weights satisfy $\sum_i w_i = 1$, the variance of the weighted estimator is:*

$$\text{Var}[\hat{S}_\ell] = \sum_i w_i^2 \text{Var}[g_\ell] = \sigma^2 \sum_i w_i^2 \tag{14}$$

*For confidence weighting with sample $i$ having $c_i = \bar{c} + \delta$ (where $\delta > 0$ indicates easier than average):*

$$w_i^2 \leq \frac{1}{|\mathcal{B}|^2} \exp(-2\gamma\delta) \tag{15}$$

*For such an easy sample, its squared weight contribution relative to uniform weighting ($w_i = 1/|\mathcal{B}|$) is bounded by $\exp(-2\gamma\delta)$. For $\gamma = 2$ and $\delta = 0.3$, this gives a suppression factor $\exp(-1.2) \approx 0.30$ for that sample's contribution.*

*Proof.* We derive the variance formula $\text{Var}[\sum_i w_i g_{\ell,i}] = \sum_i w_i^2 \text{Var}[g_\ell]$ from first principles using the definition of variance and properties of i.i.d. samples.

By definition:

$$\text{Var}\left[\sum_i w_i g_{\ell,i}\right] = \mathbb{E}\left[\left(\sum_i w_i g_{\ell,i}\right)^2\right] - \left(\mathbb{E}\left[\sum_i w_i g_{\ell,i}\right]\right)^2 \tag{16}$$

Since each $g_{\ell,i}$ has mean $\mu$ and $\sum_i w_i = 1$:

$$\mathbb{E}\left[\sum_i w_i g_{\ell,i}\right] = \sum_i w_i \mathbb{E}[g_{\ell,i}] = \sum_i w_i \mu = \mu \tag{17}$$

Expanding $(\sum_i w_i g_{\ell,i})^2$:

$$\begin{aligned}
\mathbb{E}\left[\left(\sum_i w_i g_{\ell,i}\right)^2\right] &= \mathbb{E}\left[\sum_i w_i^2 g_{\ell,i}^2 + \sum_{i \neq j} w_i w_j g_{\ell,i} g_{\ell,j}\right] \\
&= \sum_i w_i^2 \mathbb{E}[g_{\ell,i}^2] + \sum_{i \neq j} w_i w_j \mathbb{E}[g_{\ell,i}]\mathbb{E}[g_{\ell,j}]
\end{aligned} \tag{18}$$

where the second equality uses independence: $\mathbb{E}[g_{\ell,i} g_{\ell,j}] = \mathbb{E}[g_{\ell,i}]\mathbb{E}[g_{\ell,j}]$ for $i \neq j$.

The cross-term sum becomes:

$$\sum_{i \neq j} w_i w_j \mathbb{E}[g_{\ell,i}]\mathbb{E}[g_{\ell,j}] = \mu^2 \sum_{i \neq j} w_i w_j = \mu^2\left(\left(\sum_i w_i\right)^2 - \sum_i w_i^2\right) = \mu^2\left(1 - \sum_i w_i^2\right) \tag{19}$$

Substituting into the variance formula:

$$\begin{aligned}
\mathrm{Var}\left[\sum_i w_i g_{\ell,i}\right] &= \sum_i w_i^2 \mathbb{E}[g_{\ell,i}^2] + \mu^2\left(1 - \sum_i w_i^2\right) - \mu^2 \\
&= \sum_i w_i^2 \mathbb{E}[g_{\ell,i}^2] - \mu^2 \sum_i w_i^2 \\
&= \sum_i w_i^2 \left(\mathbb{E}[g_{\ell,i}^2] - \mu^2\right) \\
&= \sum_i w_i^2 \mathrm{Var}[g_{\ell,i}] = \sigma^2 \sum_i w_i^2
\end{aligned} \tag{20}$$

where the last equality uses $\mathrm{Var}[g_{\ell,i}] = \sigma^2$ for all $i$ (identically distributed).

From Proposition A.1, for $c_i = \bar{c} + \delta$:

$$w_i \leq \frac{1}{|\mathcal{B}|}\exp(-\gamma\delta) \implies w_i^2 \leq \frac{1}{|\mathcal{B}|^2}\exp(-2\gamma\delta) \tag{21}$$

Under uniform weighting, $w_i = 1/|\mathcal{B}|$, so:

$$\mathrm{Var}_{\text{uniform}} = \sum_i \frac{1}{|\mathcal{B}|^2}\mathrm{Var}[g_\ell] = \frac{1}{|\mathcal{B}|}\mathrm{Var}[g_\ell] \tag{22}$$

The squared-weight contribution ratio for easy samples is:

$$\frac{w_i^2}{1/|\mathcal{B}|^2} \leq \exp(-2\gamma\delta) \tag{23}$$

$\square$

This simplified analysis suggests that confidence weighting can reduce the influence of confidently solved samples in the importance estimator. It should be viewed as motivation for the probing design; the empirical ablations in Section 4.5.1 evaluate its practical effect.

**Taylor Saliency: First-Order Approximation Analysis.** We justify the choice of $S_\ell = \sum_{i,j} |W_\ell^{(i,j)} G_\ell^{(i,j)}|$ as the importance metric by analyzing proportional perturbations.

**Theorem A.3** (Taylor saliency as sensitivity indicator). *Consider proportional perturbations $\Delta W_\ell = -\alpha W_\ell$ that shrink all weights uniformly by factor $\alpha \in [0,1]$. By Taylor's theorem with bounded Hessian $\|H_\ell\|_{op} \leq M$, the loss change satisfies:*

$$\mathcal{L}(W_\ell - \alpha W_\ell) - \mathcal{L}(W_\ell) = -\alpha \langle G_\ell, W_\ell \rangle + O(\alpha^2 \|W_\ell\|_F^2) \tag{24}$$

*where the first-order term is:*

$$\Delta \mathcal{L}_\ell^{(1)} = -\alpha \sum_{i,j} G_\ell^{(i,j)} W_\ell^{(i,j)} \tag{25}$$

*Taking absolute values and averaging over batches, we obtain the Taylor saliency:*

$$S_\ell = \frac{1}{T} \sum_{t=1}^{T} \sum_{i,j} |W_\ell^{(i,j)} G_\ell^{(i,j)}| = \frac{1}{T} \sum_{t=1}^{T} \|W_\ell \odot G_\ell\|_1 \tag{26}$$

*which bounds the expected absolute loss change:*

$$\mathbb{E}_t[|\Delta \mathcal{L}_\ell^{(1)}|] \leq \alpha \cdot S_\ell \tag{27}$$

*Proof.* The first-order Taylor expansion gives:

$$\mathcal{L}(W_\ell - \alpha W_\ell) - \mathcal{L}(W_\ell) \approx \langle G_\ell, -\alpha W_\ell \rangle = -\alpha \sum_{i,j} G_\ell^{(i,j)} W_\ell^{(i,j)} \tag{28}$$

Taking absolute values and applying triangle inequality:

$$|\Delta \mathcal{L}_\ell^{(1)}| = \alpha \left| \sum_{i,j} G_\ell^{(i,j)} W_\ell^{(i,j)} \right| \leq \alpha \sum_{i,j} |G_\ell^{(i,j)} W_\ell^{(i,j)}| = \alpha \|W_\ell \odot G_\ell\|_1 \tag{29}$$

Averaging over $T$ batches yields the bound. The second-order error is $O(\alpha^2 M \|W_\ell\|_F^2)$ by standard Taylor remainder bounds. $\square$

The metric $S_\ell = \|W \odot G\|_1$ captures sensitivity to proportional perturbations, which naturally arise in low-rank adaptation where we constrain the update magnitude. Layers with high $S_\ell$ are sensitive to parameter changes and should receive higher ranks. The bound is conservative (tight when gradients have consistent sign), which is desirable—we prefer to over-allocate to important layers rather than under-allocate.

**Logarithmic Smoothing: Concentration Analysis.** We analyze why raw importance scores lead to excessive budget concentration and how logarithmic smoothing mitigates this issue.

In practice, we observe that raw Taylor saliency scores $\{S_\ell\}$ exhibit heavy-tailed distributions: a small fraction of layers have disproportionately large scores. Direct proportional allocation $r_\ell \propto S_\ell$ leads to pathological concentration where most budget is allocated to very few layers, leaving most layers at minimum rank.

**Proposition A.4** (Logarithmic smoothing effect). *Logarithmic smoothing $\tilde{s}_\ell = \log(1 + S_\ell)$ compresses the dynamic range of scores. For scores spanning multiple orders of magnitude (e.g., $S_{\max}/S_{\min} = K$ where $K \gg 1$), the allocation ratio is transformed as follows.*

*Without smoothing, the rank ratio is:*

$$\frac{r_{\max}}{r_{\min}} \propto \frac{S_{\max}}{S_{\min}} = K \tag{30}$$

*With smoothing, the rank ratio becomes:*

$$\frac{\tilde{r}_{\max}}{\tilde{r}_{\min}} \propto \frac{\log(1 + S_{\max})}{\log(1 + S_{\min})} = \frac{\log(1 + K S_{\min})}{\log(1 + S_{\min})} \tag{31}$$

*For large $K$ and $S_{\min} \geq 1$, this ratio grows only logarithmically:*

$$\frac{\log(1 + KS_{\min})}{\log(1 + S_{\min})} \approx \frac{\log(KS_{\min})}{\log S_{\min}} = 1 + \frac{\log K}{\log S_{\min}} = O(\log K) \tag{32}$$

*compared to the linear growth $O(K)$ without smoothing.*

*Proof.* For $S_{\min} \geq 1$ and $K \gg 1$, we have $KS_{\min} \gg 1$, so:

$$\log(1 + KS_{\min}) \approx \log(KS_{\min}) = \log K + \log S_{\min} \tag{33}$$

and similarly:

$$\log(1 + S_{\min}) \approx \log S_{\min} \tag{34}$$

Therefore:

$$\frac{\log(1 + KS_{\min})}{\log(1 + S_{\min})} \approx \frac{\log K + \log S_{\min}}{\log S_{\min}} = 1 + \frac{\log K}{\log S_{\min}} \tag{35}$$

For fixed $S_{\min}$, this grows as $O(\log K)$ when $K \to \infty$, whereas the unsmoothed ratio grows as $O(K)$. $\square$

The precise quantitative effect depends on the actual score distribution, which varies by model and task. Our ablation experiments (Section 4.4.1) empirically validate that logarithmic smoothing significantly improves performance over both uniform allocation and raw proportional allocation, confirming its effectiveness in practice.

**Rank Allocation: Marginal Utility Analysis.** We justify the proportional allocation rule $r_\ell \propto \tilde{s}_\ell$ by analyzing the marginal utility of rank allocation under diminishing returns.

**Proposition A.5** (Linear allocation from logarithmic utility). *Consider allocating total rank budget $R_{tot}$ to $L$ layers with importance scores $\{s_\ell\}$ to maximize aggregate utility:*

$$\max_{\{r_\ell\}} \sum_{\ell=1}^{L} s_\ell \cdot u(r_\ell) \quad s.t. \quad \sum_{\ell=1}^{L} r_\ell = R_{tot}, \quad r_\ell \geq r_{\min} \tag{36}$$

*where $u : \mathbb{R}_+ \to \mathbb{R}_+$ is a concave utility function. Concavity means $u''(r) < 0$, so the marginal utility $u'(r)$ is positive but decreasing (diminishing marginal returns).*

*For the logarithmic utility $u(r) = \log(1 + r)$, the optimal allocation is exactly linear in importance scores:*

$$r_\ell^* = r_{\min} + (R_{tot} - L \cdot r_{\min}) \cdot \frac{s_\ell}{\sum_k s_k} \tag{37}$$

*Proof.* By the Lagrange multiplier method, at optimum we have:

$$\frac{\partial}{\partial r_\ell} \left[ \sum_\ell s_\ell u(r_\ell) - \lambda \sum_\ell r_\ell \right] = 0 \implies s_\ell u'(r_\ell^*) = \lambda \quad \text{for all } \ell \tag{38}$$

For logarithmic utility $u(r) = \log(1 + r)$, we have:

- $u'(r) = \frac{1}{1+r} > 0$ (positive marginal utility)

- $u''(r) = -\frac{1}{(1+r)^2} < 0$ (diminishing returns, i.e., concave)

The Lagrange condition becomes:

$$s_\ell \cdot \frac{1}{1 + r_\ell^*} = \lambda \implies 1 + r_\ell^* = \frac{s_\ell}{\lambda} \implies r_\ell^* = \frac{s_\ell}{\lambda} - 1 \tag{39}$$

To find $\lambda$, use the budget constraint $\sum_\ell r_\ell^* = R_{\text{tot}}$:

$$\sum_\ell \left( \frac{s_\ell}{\lambda} - 1 \right) = R_{\text{tot}} \implies \frac{\sum_\ell s_\ell}{\lambda} = R_{\text{tot}} + L \implies \lambda = \frac{\sum_\ell s_\ell}{R_{\text{tot}} + L} \tag{40}$$

Substituting back:

$$r_\ell^* = \frac{s_\ell}{\lambda} - 1 = s_\ell \cdot \frac{R_{\text{tot}} + L}{\sum_k s_k} - 1 = (R_{\text{tot}} + L) \cdot \frac{s_\ell}{\sum_k s_k} - 1 \tag{41}$$

To incorporate minimum rank $r_{\min}$, we allocate the "excess budget" $R_{\text{tot}} - L \cdot r_{\min}$ proportionally:

$$r_\ell^* = r_{\min} + (R_{\text{tot}} - L \cdot r_{\min}) \cdot \frac{s_\ell}{\sum_k s_k} \tag{42}$$

This is exactly the linear allocation rule used in CSPLoRA. $\qquad \square$

The choice $u(r) = \log(1 + r)$ is natural for rank allocation because: (1) Diminishing returns—adding rank from 2 to 4 is more valuable than from 16 to 18, consistent with the observation that expressiveness gains saturate at higher ranks; (2) Exact linear allocation—unlike power-law utilities $u(r) = r^\beta$ which only yield approximately linear allocation, logarithmic utility gives exactly linear allocation; (3) Well-defined at zero—$u(0) = 0$ and $u'(0) = 1$, providing a natural reference point.

## B. Expanded Experiments and Ablations

This section provides additional experimental details and ablation studies.

### B.1. Top-$p$ Coverage ($\rho$)

The coverage parameter $\rho$ controls what fraction of layers receive non-minimum ranks. Table 11 shows the detailed per-task breakdown. While performance is relatively stable across different $\rho$ values, $\rho = 1.0$ (full coverage) achieves the highest average accuracy (73.51), outperforming other settings. We adopt $\rho = 1.0$ throughout our experiments for its robustness and simplicity.

Table 11. Ablation on top-$p$ coverage $\rho$ (LLaMA-2-7B, 5-task commonsense subset).

| $\rho$ | BoolQ | HeSw | PIQA | ARC-C | WiGr | Avg |
|---|---|---|---|---|---|---|
| 0.7 | 68.28±0.08 | 78.79±2.06 | 79.82±0.43 | 63.99±0.66 | 76.27±0.39 | 73.43 |
| 0.8 | 68.44±0.57 | 78.99±2.46 | 79.80±0.40 | 63.68±0.83 | 76.45±0.32 | 73.47 |
| 0.9 | 68.39±0.54 | 78.51±2.68 | 79.63±0.14 | 63.99±0.54 | 76.59±0.24 | 73.42 |
| 0.95 | 68.60±0.27 | 77.90±1.31 | 79.98±0.31 | 64.05±0.26 | 76.37±0.43 | 73.38 |
| 1.0 (Default) | 68.50±0.35 | 78.48±3.85 | 79.85±0.22 | 64.14±0.38 | 76.56±0.22 | **73.51** |

### B.2. Per-Task Performance Breakdown

This section provides detailed per-task breakdowns of the experiments reported in the main text, which only show average scores due to space constraints.

Table 12. Full breakdown of confidence weighting $\gamma$ ablation (LLaMA-2-7B, 5-task commonsense subset).

| $\gamma$ | BoolQ | HeSw | PIQA | ARC-C | WiGr | Avg |
|---|---|---|---|---|---|---|
| 0.0 (Uniform) | 68.53±0.63 | 78.13±2.92 | 79.78±0.34 | 63.59±0.50 | 76.64±0.62 | 73.33 |
| 1.0 | 68.34±0.63 | 78.55±0.55 | 79.87±0.12 | 63.59±0.16 | 76.80±0.50 | 73.43 |
| 2.0 (Default) | 68.50±0.35 | 78.48±3.85 | 79.85±0.22 | 64.14±0.38 | 76.56±0.22 | **73.51** |
| 4.0 | 68.53±0.81 | 78.63±0.83 | 79.96±0.05 | 63.59±0.38 | 76.71±0.30 | 73.48 |

**Confidence Weighting $\gamma$ Ablation.** Table 12 provides the full 5-task breakdown for the confidence weighting ablation presented in Section 4.5.1.

**Structural Component Ablation.** Table 13 provides the full 5-task breakdown for the structural component ablation presented in Section 4.4.1.

*Table 13.* Full breakdown of structural component ablation (LLaMA-2-7B, 5-task commonsense subset).

| Configuration | BoolQ | HeSw | PIQA | ARC-C | WiGr | Avg |
|---|---|---|---|---|---|---|
| CSPLoRA (Full) | 68.50±0.35 | 78.48±3.85 | 79.85±0.22 | 64.14±0.38 | 76.56±0.22 | **73.51** |
| w/o Calibration | 68.28±0.40 | 77.67±3.69 | 79.76±0.19 | 63.77±0.84 | 76.43±0.13 | 73.18 |
| w/o Smoothing | 68.37±0.57 | 77.09±0.40 | 79.98±0.43 | 63.42±0.73 | 76.53±0.33 | 73.08 |
| w/o Both | 68.14±0.45 | 76.82±2.65 | 79.89±0.20 | 63.74±0.91 | 76.30±0.16 | 72.98 |

**Importance Metric Comparison.** Table 14 provides the full 5-task breakdown for the importance metric comparison presented in Section 4.4.2.

*Table 14.* Full breakdown of importance metric comparison (LLaMA-2-7B, 5-task commonsense subset).

| Metric | BoolQ | HeSw | PIQA | ARC-C | WiGr | Avg |
|---|---|---|---|---|---|---|
| Fisher ($\|G\|_F^2$) | 68.28±0.47 | 78.72±3.77 | 79.83±0.30 | 63.74±0.39 | 76.30±0.45 | 73.37 |
| Mean Taylor | 68.44±0.62 | 78.71±3.49 | 79.94±0.27 | 63.68±0.53 | 76.56±0.53 | 73.47 |
| Sum Taylor (Ours) | 68.50±0.35 | 78.48±3.85 | 79.85±0.22 | 64.14±0.38 | 76.56±0.22 | **73.51** |

**Cross-Budget Reuse.** Table 15 provides the full 5-task breakdown for the cross-budget reuse experiment presented in Section 4.6.1.

*Table 15.* Full breakdown of cross-budget reuse (LLaMA-2-7B, 5-task commonsense subset). All CSPLoRA results use importance scores from a single probing run.

| Rank | Method | BoolQ | HeSw | PIQA | ARC-C | WiGr | Avg |
|---|---|---|---|---|---|---|---|
| $r = 4$ | LoRA | 66.44±0.82 | 72.33±4.00 | 79.42±0.48 | 62.32±0.22 | 74.74±0.11 | 71.05 |
| | CSPLoRA | 66.43±1.64 | 74.15±1.92 | 79.36±0.39 | 62.68±0.43 | 74.77±0.10 | **71.48** |
| $r = 8$ | LoRA | 67.43±0.70 | 75.12±4.99 | 79.13±0.58 | 62.60±0.26 | 74.82±0.57 | 71.82 |
| | CSPLoRA | 68.50±0.35 | 78.48±3.85 | 79.85±0.22 | 64.14±0.38 | 76.56±0.22 | **73.51** |
| $r = 16$ | LoRA | 69.67±0.57 | 81.61±3.30 | 80.59±0.54 | 65.24±0.60 | 77.90±0.22 | 75.01 |
| | CSPLoRA | 69.58±0.31 | 82.67±3.99 | 80.56±0.69 | 65.36±0.42 | 77.56±0.44 | **75.15** |

# C. Additional Diagnostics on RTX 5090

The following diagnostics further examine parameter budgets and robustness under an RTX 5090 environment. We report them separately from the main A100/A800 experiments and compare methods only within each table.

## C.1. Parameter Audit

Table 16 reports a single-run parameter audit on the 5-task commonsense subset. The comparison confirms that CSPLoRA does not rely on a larger trainable-parameter budget than uniform LoRA in this diagnostic setting.

*Table 16.* Parameter audit on RTX 5090 for the 5-task commonsense subset.

| Method | Params | BoolQ | HeSw | PIQA | ARC-C | WiGr | Avg |
|---|---|---|---|---|---|---|---|
| LoRA | 19.99M | 67.89 | 78.06 | 80.47 | 63.82 | 76.56 | 73.36 |
| CSPLoRA | 19.98M | 68.53 | 81.83 | 79.92 | 63.31 | 76.95 | **74.11** |

## C.2. Adaptive Confidence Weighting

Table 17 provides the per-task breakdown of the RTX 5090 confidence-weighting diagnostic summarized in Table 7. In this within-environment comparison, adaptive weighting improves the average over both uniform probing and the

fixed-temperature variant.

Table 17. Per-task confidence-weighting diagnostic on RTX 5090.

| Setting | BoolQ | HeSw | PIQA | ARC-C | WiGr | Avg |
|---------|-------|------|------|-------|------|-----|
| LoRA | 68.10±0.30 | 79.71±2.32 | 80.22±0.35 | 63.48±0.48 | 76.44±0.17 | 73.59 |
| 0.0 (Uniform) | 68.32±0.22 | 81.33±3.11 | 80.14±0.31 | 63.14±0.00 | 76.80±0.11 | 73.95 |
| 2.0 (Fixed) | 68.30±0.15 | 81.27±1.98 | 80.17±0.35 | 63.48±0.12 | 76.99±0.61 | 74.04 |
| Adaptive | 68.44±0.13 | 82.62±1.12 | 80.30±0.54 | 63.87±0.78 | 76.76±0.28 | **74.40** |

## C.3. Minimum-Rank Sensitivity

Table 18 evaluates the minimum-rank constraint. The results indicate that $r_{\min}$ controls the flexibility of the final allocation: moderate values remain competitive, while changing the constraint can shift task-level performance.

Table 18. Minimum-rank sensitivity on RTX 5090 for the 5-task commonsense subset.

| $r_{\min}$ | BoolQ | HeSw | PIQA | ARC-C | WiGr | Avg |
|------------|-------|------|------|-------|------|-----|
| 0 | 69.02 | 78.07 | 80.25 | 64.08 | 77.43 | 73.77 |
| 1 | 68.26 | 80.88 | 80.47 | 63.48 | 76.16 | 73.85 |
| 2 | 68.53 | 81.83 | 79.92 | 63.31 | 76.95 | **74.11** |
| 4 | 68.47 | 79.36 | 79.76 | 63.48 | 76.48 | 73.51 |

## C.4. Encoder-Decoder Diagnostic on T5

Table 19 reports an additional T5-base diagnostic on GLUE. This result provides complementary evidence beyond the decoder-only and encoder-only settings in the main experiments, while broader encoder-decoder validation remains future work.

Table 19. Additional T5-base GLUE diagnostic on RTX 5090.

| Method | MNLI | QQP | QNLI | SST-2 | CoLA | MRPC | RTE | STS-B | Avg |
|--------|------|-----|------|-------|------|------|-----|-------|-----|
| LoRA | 85.71±0.01 | 88.78±0.01 | 93.14±0.05 | 94.09±0.08 | 51.15±1.33 | 87.11±0.55 | 70.76±0.00 | 89.70±0.06 | 82.55 |
| CSPLoRA | 85.87±0.00 | 88.84±0.02 | 93.22±0.04 | 94.32±0.08 | 52.11±1.15 | 88.18±0.43 | 70.58±0.26 | 89.76±0.01 | **82.86** |

# D. Hyperparameters

This appendix provides detailed hyperparameters for the main experiments and structural planning procedure.

## D.1. Training Configuration

Table 20 summarizes the hyperparameters used across the main experiments. For LLaMA models, we use learning rate 5e-5 (LLaMA-2) or 1e-4 (LLaMA-3.1) for LoRA methods. For RoBERTa-large on GLUE, we follow standard practice with 10 epochs on small datasets (MRPC, RTE, STS-B) and 3 epochs on larger datasets.

Table 20. Hyperparameters used in experiments.

| Model | Task | LR | Batch | Epochs | Seq Len |
|-------|------|----|-------|--------|---------|
| LLaMA-2-7B | Commonsense | 5e-5 | 32 | 1 | 256 |
| LLaMA-3.1-8B | Commonsense | 1e-4 | 32 | 1 | 256 |
| LLaMA-2-7B | Arithmetic | 5e-5 | 16 | 1 | 512 |
| LLaMA-3.1-8B | Arithmetic | 1e-4 | 32 | 1 | 512 |
| RoBERTa-large | GLUE | 5e-5 | 32 | 3/10 | 512 |

All LoRA experiments use rank $r = 8$, alpha $\alpha = 16$, and dropout 0.05. We use AdamW optimizer with cosine learning rate

schedule and warmup ratio 0.02 (0.06 for GLUE). Weight decay is set to 0 for LLaMA-2, 5e-4 for LLaMA-3.1 commonsense, 0.1 for LLaMA-3.1 arithmetic and GLUE.

### D.2. Probe Configuration

Table 21 shows CSPLoRA-specific hyperparameters for structural planning. Main experiments use a fixed confidence weight $\gamma = 2.0$, top-$p$ coverage $\rho = 1.0$ to allocate budget to all layers, and minimum rank $r_{\min} = 2$ to ensure basic expressivity. The RTX 5090 diagnostics additionally evaluate the adaptive confidence-weighting strategy described in Section 3.2. The maximum rank is set to $r_{\max} = 32$, and probing runs for up to 200 steps using grouped low-memory mode to reduce peak memory usage. As shown in Section 4.3.2, importance scores typically stabilize within 50–100 steps, so the actual number of probe steps is often lower.

*Table 21.* CSPLoRA probe hyperparameters.

| Parameter | Value |
|---|---|
| Fixed confidence weight ($\gamma$) | 2.0 |
| Adaptive gamma scale ($\gamma_s$) | 1.0 |
| Top-$p$ coverage ($\rho$) | 1.0 |
| Minimum rank ($r_{\min}$) | 2 |
| Maximum rank ($r_{\max}$) | 32 |
| Max Probe steps | 200 |

### D.3. LoRA Configuration

All LoRA variants use the same base hyperparameters across experiments: rank $r = 8$, alpha 16, and dropout 0.05. For LLaMA models, we apply LoRA to all linear projections in attention and feed-forward layers (q_proj, k_proj, v_proj, o_proj, gate_proj, up_proj, down_proj). For RoBERTa, we target query, value, attention output dense, and FFN output dense layers.

## E. Algorithm

Algorithm 1 presents the complete CSPLoRA procedure, consisting of three stages: confidence-guided probing, Taylor saliency computation, and proportional rank allocation.

---

**Algorithm 1** CSPLoRA: Confidence-Guided Probing and Scale-Invariant Rank Allocation

---

**Require:** Pre-trained model with weights $\{W_\ell\}_{\ell=1}^L$, probe dataset $\mathcal{D}_{\text{probe}}$
**Require:** Hyperparameters: confidence-weighting strategy, $\rho$ (coverage), $r_{\min}$ (minimum rank), $R_{\text{tot}}$ (total rank budget)
**Ensure:** Rank allocation $\{r_\ell\}_{\ell=1}^L$
  1: **// Stage 1: Confidence-guided Probing**
  2: Initialize layer-wise importance accumulators $S_\ell \leftarrow 0$ for all layers $\ell$
  3: **for** each batch $\mathcal{B} \subseteq \mathcal{D}_{\text{probe}}$ (randomly sampled) **do**
  4:     Compute confidence $c_i = p_\theta(y_i|x_i)$ and difficulty $d_i = 1 - c_i$ for each sample $i \in \mathcal{B}$
  5:     Set $\gamma_{\text{eff}}$ from the chosen strategy: fixed $\gamma$, adaptive $\gamma_s/\text{std}(\{d_i\})$, or 0 for uniform weighting
  6:     Compute weights $w_i = \text{softmax}_i(\gamma_{\text{eff}} d_i)$, using centered difficulties for the adaptive variant
  7:     Compute weighted loss $\mathcal{L}_w = \sum_{i \in \mathcal{B}} w_i \cdot \ell(f_\theta(x_i), y_i)$
  8:     Backpropagate to obtain gradients $G_\ell = \nabla_{W_\ell} \mathcal{L}_w$
  9:     **// Stage 2: Taylor Saliency Accumulation**
 10:     **for** each layer $\ell = 1, \dots, L$ **do**
 11:         $S_\ell \leftarrow S_\ell + \|W_\ell \odot G_\ell\|_1$
 12:     **end for**
 13: **end for**
 14: Normalize: $S_\ell \leftarrow S_\ell/T$ where $T$ is the number of batches
 15: **// Stage 3: Scale-Invariant Rank Allocation**
 16: Apply score calibration across target-module families
 17: **// Log smoothing prevents pathological concentration (Prop. A.4)**
 18: **for** each layer $\ell = 1, \dots, L$ **do**
 19:     Apply smoothing: $\tilde{s}_\ell = \log(1 + S_\ell)$
 20: **end for**
 21: Sort layers by $\tilde{s}_\ell$ in descending order
 22: Select top-$\rho$ fraction: $\mathcal{L}_{\text{active}} = \{\ell : \text{rank}(\tilde{s}_\ell) \leq \lceil \rho \cdot L \rceil\}$
 23: Compute normalization: $Z = \sum_{\ell \in \mathcal{L}_{\text{active}}} \tilde{s}_\ell$
 24: **// Proportional allocation (Prop. A.5)**
 25: **for** each layer $\ell = 1, \dots, L$ **do**
 26:     **if** $\ell \in \mathcal{L}_{\text{active}}$ **then**
 27:         $r_\ell = r_{\min} + \lfloor (R_{\text{tot}} - L \cdot r_{\min}) \cdot \tilde{s}_\ell/Z \rfloor$
 28:     **else**
 29:         $r_\ell = r_{\min}$
 30:     **end if**
 31: **end for**
 32: **return** $\{r_\ell\}_{\ell=1}^L$

---

