# OpenReview forum: "CSPLoRA: Confidence-Guided Structure Planning for Low-Rank Adaptation"
_ICML.cc/2026/Conference — ICML 2026 regular_

### Official Review · Reviewer_XMwX · 2026-03-12

**Soundness:** 2
**Presentation:** 3
**Significance:** 2
**Originality:** 2
**Overall Recommendation:** 3
**Confidence:** 5

**Summary:**

This paper proposes CSPLoRA, a decoupled structural planning framework for LoRA that determines rank allocation across model modules before training begins. The central contribution is a confidence-guided probing mechanism that upweights hard samples (low model confidence) when computing Taylor saliency scores, on the premise that these samples yield more informative gradients for identifying which modules most need adaptation. Combined with logarithmic smoothing and a proportional rank allocation rule, the method produces structural priors that are claimed to be reusable across different rank budgets and LoRA backends - the "probe once, deploy everywhere" paradigm. Experiments are conducted on GLUE, commonsense reasoning, and arithmetic reasoning tasks using LLaMA-2-7B, LLaMA-3.1-8B, and RoBERTa-large.

**Compliance With Llm Reviewing Policy:**

Affirmed.

**Key Questions For Authors:**

+ Confidence weighting gains are marginal. The ablation in Table 5 shows only a 0.18-point improvement from $\gamma=0$ to $\gamma=2.0$. Can the authors provide a more direct empirical validation that confidence weighting improves the signal-to-noise ratio of importance estimates - for example, by measuring gradient variance or importance score stability across seeds with and without weighting? If so, would this change the evaluation?

+ CSPLoRA trails GoRA by 2.51 points on LLaMA-2-7B commonsense reasoning (74.74 vs. 77.25). Can the authors explain under what practical conditions CSPLoRA would be preferable to GoRA, beyond computational reusability?

+ Generalization to encoder-decoder models. All evaluations use decoder-only or encoder-only architectures. Do the authors have any evidence or theoretical reason to believe CSPLoRA's structural priors would transfer effectively to encoder-decoder settings such as T5 or BART?

**Limitations:**

Yes

**Strengths And Weaknesses:**

**A. Strengths**

**1. Soundness**

+ The theoretical analysis is generally coherent. Theorem A.2 provides a reasonable (if idealized) variance reduction argument which supports confidence reweighting. Also, Proposition A.5 formally justifies the usage of proportional allocation under logarithmic utility and diminishing marginal returns.

+ The scale-invariance property (Eq. 8) is well-motivated and correctly derived

+ The paper has thorough and internally consistent ablations: smoothing, normalization, $\gamma$, coverage $\rho$, and importance metric are all studied.

**2. Presentation**

+ The paper is clearly written and well-structured. The pipeline presented in the main figure is also clear.

+ The key contributions, including data-centric weighting and metric-centric prior work, are presented clearly and early

**3. Significance**

+ The "probe once, deploy everywhere" property has genuine practical value. A single probing run that transfers across rank budgets meaningfully reduces experimentation cost compared to prior adaptive rank allocation LoRA-based methods that require full retraining per configuration.

+ Some parameter efficiency results (Table 8) are favorable: CSPLoRA uses fewer parameters than AdaLoRA while matching or exceeding it in performance.

**4. Originality**

+ The data-centric framing that reweights probe samples by prediction confidence for structural planning is a novel angle that complements existing metric-centric approaches like GoRA.

+ The combination of confidence-weighted probing with scale-invariant proportional allocation, explicitly targeting cross-budget and cross-backend reusability, has not been systematically explored in prior works.

**B. Weaknesses**

**1. Soundness**

+ The practical implication of Theorem A.2 - that confidence weighting meaningfully improves the signal-to-noise ratio of importance estimates - is not empirically verified. No direct measurement of gradient variance or estimation quality is provided, and the downstream performance gap between $\gamma=0$ and $\gamma=2.0$ ($0.18$ points, Table 5) is too small to constitute strong validation of the theoretical motivation.

+ Proposition A.5 provides a clean and formally correct justification for proportional allocation under logarithmic utility, though it is worth noting that the result is somewhat tailored to that specific utility function - other concave utilities would yield different allocation rules. Nevertheless, this is a minor concern, as the allocation rule is well-validated empirically, but the proposition is better understood as confirmatory theory rather than a derivation from first principles.

**2. Presentation**

+ The abstract and introduction overstate the role of confidence weighting relative to what the ablations actually support. The gains from confidence weighting ($\gamma=2.0$ vs. $\gamma=0.0$) are $0.18$ points, which is within the noise of most comparisons, yet the mechanism is framed as a primary contribution throughout.

**3. Significance**

+ The empirical improvements over uniform LoRA are frequently within one standard deviation. On LLaMA-3.1-8B, CSP-LoRA+ $(84.44)$ underperforms the LoRA+ baseline $(84.69)$, and gains across other backends on this model are marginal at best.

+ CSPLoRA consistently and substantially trails GoRA on the primary benchmark (LLaMA-2-7B commonsense: $74.74$ vs. $77.25$ with the LoRA backend). The paper frames these methods as complementary rather than competing, but the gap is large enough to require a more direct explanation of when and why CSPLoRA is preferable.

+ Evaluation is limited to decoder-only and encoder-only architectures. The absence of encoder-decoder tasks (e.g., summarization, translation), settings where LoRA is widely deployed, limits the generalizability of the conclusions.

**4. Originality**

+ The individual components (Taylor saliency, logarithmic smoothing, confidence-weighted loss) are each borrowed or closely adapted from prior work. The novelty lies in their combination and the reusability framing, which is incremental rather than transformative.

---

> ### Author Rebuttal · Authors · 2026-03-30
>
> We thank the reviewer for these points. The main concerns are `(i)` whether the empirical support for confidence weighting is sufficient, `(ii)` what the practical value of `CSPLoRA` is relative to GoRA, and `(iii)` whether encoder-decoder evidence is missing. We address them with `γ` and `T5` results and will narrow the corresponding wording in the paper.
>
> ### 1 On the gain from confidence weighting and its empirical support
>
> We understand the reviewer’s concern. In the current paper, the gain from `γ=0` to fixed `γ=2.0` is better viewed as a stable but moderate empirical gain, and should not be phrased as an overly strong “significantly improves signal-to-noise ratio” claim. Accordingly, we view `confidence weighting` as an empirical mechanism that improves probing quality, rather than the sole source of the method’s effectiveness.
>
> The key question, however, is not only whether fixed `γ` is useful, but also whether the weighting strength can be set more reasonably. Fixed `γ` is simple but global, while the difficulty dispersion of local probe subsets is not the same throughout probing: when it is narrow, fixed `γ` may not sufficiently separate hard and easy samples; when it is wide, it may over-concentrate weight on a few samples. Based on this observation, and without changing the original confidence-aware probing framework, we further evaluate a adaptive`γ` variant which is more flexible. We still define sample difficulty as `1 - confidence`, but instead of using a single fixed constant, we automatically adjust the effective weighting strength according to the difficulty dispersion observed during probing. The goal is not to turn `γ` into another hyperparameter requiring task-by-task manual search, but to make the weighting in probing better match local task signals. Based on this, we further conduct a multi-seed comparison on `5-task`:
>
> |Setting|BoolQ|HeSw|PIQA|ARC-C|WiGr|Avg|
> |---|---:|---:|---:|---:|---:|---:|
> |LoRA|68.10±0.30|79.71±2.32|80.22±0.35|63.48±0.48|76.44±0.17|73.59|
> |`γ=0`|68.32±0.22|81.33±3.11|80.14±0.31|63.14±0.00|76.80±0.11|73.95|
> |`γ=2`|68.30±0.15|81.27±1.98|80.17±0.35|63.48±0.12|76.99±0.61|74.04|
> |Adaptive|68.44±0.13|82.62±1.12|80.30±0.54|63.87±0.78|76.76±0.28|**74.40**|
>
> These results indicate a consistent and meaningful practical benefit for confidence-aware probing. Adaptive `γ` achieves the best average result, further suggesting that a rule adapting to the difficulty dispersion observed during probing is more flexible than a single fixed constant. Based on the current evidence, we would describe this component as a “modest but useful probing improvement.” In the revision, we will also include this adaptive `γ` variant and its empirical results as a practical option alongside fixed `γ`.
>
> ### 2 On “if GoRA is better on some main benchmarks, why use `CSPLoRA` at all?”
>
> We understand the reviewer’s comparison angle. In our view, the two methods do not operate at exactly the same level. Beyond rank allocation, `GoRA` also incorporates a specific parameter-initialization design, so it is better viewed as an integrated solution for a particular training configuration; by contrast, `CSPLoRA` mainly provides task-aware rank planning and does not restrict the downstream initialization or update strategy.
>
> More precisely, `CSPLoRA` decides how a limited rank budget is distributed across modules, thereby providing a reusable structural prior; final performance still depends on the backend, initialization, and optimization used afterwards. For this reason, `CSPLoRA` is complementary to methods such as `LoRA/LoRA+/DoRA/PiSSA`: methodologically, it is orthogonal to downstream initialization and update strategies rather than tied to one fixed recipe.
>
> This is also why we emphasize reuse in the paper. The value of `CSPLoRA` is not only a point comparison against one fixed implementation, but that it can serve as a reusable task-aware planner for different budgets and LoRA-family backends under the same task-model pair. We will make this positioning clearer in the revision and avoid presenting such comparisons as an overly absolute judgment between fixed recipes.
>
> ### 3 On the lack of encoder-decoder evidence
>
> To address the reviewer’s concern about encoder-decoder models, during the rebuttal we additionally ran `T5-base + GLUE` as a direct cross-architecture validation. The results are shown below:
>
> |Method|MNLI|QQP|QNLI|SST-2|CoLA|MRPC|RTE|STSB|Avg|
> |---|---:|---:|---:|---:|---:|---:|---:|---:|---:|
> |LoRA|85.71±0.01|88.78±0.01|93.14±0.05|94.09±0.08|51.15±1.33|87.11±0.55|70.76±0.00|89.70±0.06|82.55|
> |CSPLoRA|85.87±0.00|88.84±0.02|93.22±0.04|94.32±0.08|52.11±1.15|88.18±0.43|70.58±0.26|89.76±0.01|**82.86**|
>
> These results provide direct evidence for the encoder-decoder setting and indicate that `CSPLoRA` is not inherently tied to decoder-only architectures; it is also feasible in this setting and remains competitive overall. We will state this scope more clearly in the revision.

---

### Official Review · Reviewer_LDrB · 2026-03-13

**Soundness:** 3
**Presentation:** 3
**Significance:** 3
**Originality:** 3
**Overall Recommendation:** 4
**Confidence:** 4

**Summary:**

This paper introduces CSPLoRA, a novel framework for Low-Rank Adaptation (LoRA) that employs confidence-guided structural planning for parameter-efficient fine-tuning. The key innovation of CSPLoRA is its ability to estimate module importance prior to training, using a probe mechanism that emphasizes "hard" samples. This approach enables more effective and reusable rank allocation across different LoRA backends and rank budgets. The paper demonstrates that CSPLoRA outperforms traditional uniform rank allocation methods and adaptive rank allocation methods like AdaLoRA and GoRA on a range of tasks, including commonsense reasoning, GLUE, and arithmetic tasks.

**Compliance With Llm Reviewing Policy:**

Affirmed.

**Final Justification:**

The authors addressed most of my concerns in the rebuttal. I will keep my score.

**Key Questions For Authors:**

See above.

**Limitations:**

Yes.

**Strengths And Weaknesses:**

### Strengths
- The paper presents a technically sound approach, with well-supported claims backed by both theoretical analysis and thorough empirical results. The methods used are appropriate and well-designed, addressing the challenge of rank allocation in LoRA efficiently.
- The paper is well-structured and easy to follow. The authors effectively position their work within the existing literature, highlighting the differences and advantages of their approach.

### Weaknesses
- The writing is generally clear, but certain sections could benefit from additional clarification, particularly when explaining the assumptions behind some of the proposed methods and their theoretical foundations (Sec 3.2 and Sec 3.4).
- Could the authors provide additional experiments on newer LLM base models to demonstrate the scalability and effectiveness of CSPLoRA across different model architectures?
- Could the authors explain the trade-offs in terms of computational overhead when using CSPLoRA compared to other adaptive rank allocation methods like AdaLoRA or GoRA, especially when scaling to very large models or large-scale datasets?
- How does CSPLoRA handle tasks with highly imbalanced or noisy data? Would the proposed method still be effective in such scenarios, or could it be prone to overfitting?
- Could the authors provide more details on how the confidence parameter $\gamma$ is chosen in practice? Does it require tuning for each task, or is there a more generalized setting that works across tasks?

---

> ### Author Rebuttal · Authors · 2026-03-30
>
> We appreciate the reviewer's comments. We address the main points below.
>
> ### 1 On the method assumptions and theoretical basis in Sec. 3.2 / 3.4
>
> The paper separates task-signal extraction from structural planning: `confidence weighting` makes the saliency estimate more task-conditioned, while smoothing and allocation reduce rank collapse under a fixed budget and turn raw saliency into a more stable reusable structural prior. We will revise Sec. 3.2 / 3.4 accordingly.
>
> ### 2 On scalability to more architectures and PEFT backends
>
> To make the generalization scope more explicit, we add encoder-decoder evidence in the rebuttal and clarify the backend-level portability already shown in the main paper. Concretely, during the rebuttal period we additionally ran `T5-base + GLUE` on `RTX 5090` as a encoder-decoder validation.
>
> The results are as follows:
>
> |Method|MNLI|QQP|QNLI|SST-2|CoLA|MRPC|RTE|STSB|Avg|
> |---|---:|---:|---:|---:|---:|---:|---:|---:|---:|
> |LoRA|85.71±0.01|88.78±0.01|93.14±0.05|94.09±0.08|51.15±1.33|87.11±0.55|70.76±0.00|89.70±0.06|82.55|
> |CSPLoRA|85.87±0.00|88.84±0.02|93.22±0.04|94.32±0.08|52.11±1.15|88.18±0.43|70.58±0.26|89.76±0.01|**82.86**|
>
> This result provides direct encoder-decoder evidence that `CSPLoRA` is not tied to decoder-only models and remains competitive overall in that setting. At the backend level, the main paper already shows that the same rank prior can be reused across `LoRA/LoRA+/DoRA/PiSSA`; we will make this scope clearer and try to extend the experiments to more backends in the revision.
>
> ### 3 On the overhead and scalability trade-off compared with AdaLoRA / GoRA
>
> From the perspective of where the extra cost occurs, the additional cost of `CSPLoRA` is mainly concentrated in the initial probe/planning stage. For a fixed task-model pair, the resulting rank pattern can be saved and reused in later `LoRA/LoRA+/DoRA/PiSSA` training runs. Once the rank pattern is fixed, subsequent full training simply follows the chosen backend and introduces no additional planner-specific overhead. This is also consistent with the time breakdown reported in the main paper, where the extra time is mainly a one-time planning cost before full training and remains a relatively small portion overall. What we would like to emphasize, therefore, is reusable structural planning and amortized cost across multiple budgets or backends. We will clarify this positioning in the revision.
>
> ### 4 On the concern about noisy / imbalanced data
>
> We appreciate the concern about noisy or imbalanced data. In principle, any method that adjusts rank allocation from task signals may be affected by noise or distribution imbalance; this is not unique to `CSPLoRA`. Our weighting and smoothing are expected to provide some stabilization, but the current paper does not yet include a dedicated noisy-label or class-imbalance stress test. In the revision, we will therefore state this applicability boundary more carefully rather than phrasing it as a broader robustness claim.
>
> ### 5 On the choice of `γ` and task-wise tuning
>
> We would also like to provide a clear and more practical answer regarding the choice of `γ`. A fixed `γ` can serve as a simple default, but it may not be equally suitable for different levels of difficulty dispersion observed across local probe subsets. Therefore, in the rebuttal we additionally consider an adaptive `γ` variant. Concretely, we define sample difficulty as `1 - confidence` and adjust the weighting strength according to the difficulty dispersion observed in the probing process. Intuitively, when sample difficulty is more concentrated, the weighting becomes closer to uniform; when the difficulty spread is larger, hard samples receive stronger emphasis. The purpose of this design is not to replace fixed `γ`, but to avoid forcing a single constant to fit all local difficulty patterns observed during probing, so that the weighting strength can adjust automatically to the observed dispersion of task signals.
>
> To directly answer the reviewer’s question on how `γ` should be chosen, we ran a `5-task` comparison as follows:
>
> |Setting|BoolQ|HeSw|PIQA|ARC-C|WiGr|Avg|
> |---|---:|---:|---:|---:|---:|---:|
> |LoRA|68.10±0.30|79.71±2.32|80.22±0.35|63.48±0.48|76.44±0.17|73.59|
> |`γ=0`|68.32±0.22|81.33±3.11|80.14±0.31|63.14±0.00|76.80±0.11|73.95|
> |`γ=2`|68.30±0.15|81.27±1.98|80.17±0.35|63.48±0.12|76.99±0.61|74.04|
> |Adaptive|68.44±0.13|82.62±1.12|80.30±0.54|63.87±0.78|76.76±0.28|**74.40**|
>
> From these results, adaptive `γ` achieves the best average performance in the current supplementary experiment, suggesting that a unified adaptive rule can already provide stable gains without task-by-task manual tuning. Based on this, in the revision we will describe `γ` as a probing weight that can be handled by a unified rule, rather than as a critical hyperparameter that must be carefully tuned for each task. Fixed `γ` will remain as a simple baseline, while adaptive `γ` will be introduced as a more flexible practical option.

---

> > ### Author Rebuttal · Reviewer_LDrB · 2026-04-04
> >
> > Thank authors for the rebuttal. The clarification, as well as additional experiments address my concerns and give me more confidence. I will keep my original score.

---

### Official Review · Reviewer_Wxe6 · 2026-03-13

**Soundness:** 3
**Presentation:** 3
**Significance:** 2
**Originality:** 3
**Overall Recommendation:** 4
**Confidence:** 3

**Summary:**

The paper studies rank allocation for LoRA and proposes CSPLoRA, a structural planning method that probes module importance before training and then reuses the resulting rank structure across different rank budgets and LoRA backends. The method combines confidence-guided sample weighting, Taylor-saliency-based importance estimation, logarithmic smoothing, and proportional rank allocation. Experiments on different tasks show consistent improvements over uniform LoRA baselines and competitive performance against prior adaptive methods.

**Compliance With Llm Reviewing Policy:**

Affirmed.

**Final Justification:**

The paper studies rank allocation for LoRA and proposes CSPLoRA, a structural planning method that estimates module importance before training and then reuses the resulting rank allocation across different rank budgets and LoRA backends. The method is simple, efficient, and practical. The experimental evaluation is also solid.

My main concern was a potential fairness issue in the comparison: when the budget is measured only by the number of ranks, the actual number of trainable parameters may differ across methods, and the proposed method may tend to take more trainable parameters. In the rebuttal, the authors clarified this point and further provided experimental results under a fairer comparison setting. This addresses my main concern, and I would like to raise my score to weak accept.

**Key Questions For Authors:**

* In Section 4.4.3, the paper argues that sum Taylor is better than mean Taylor because larger layers may carry more task-relevant knowledge. However, for the same rank, larger weights also introduce more trainable parameters. Does using sum Taylor bias the method toward allocating more budget to layers where each unit of rank is itself more parameter-expensive? If so, how can we rule out the possibility that part of the gain comes simply from increasing the effective number of trainable parameters? If the authors can address this concern clearly, it would resolve one of my main reservations and would make me more positive about the paper.

* More generally, can the authors report the actual trainable parameter counts for the resulting CSPLoRA allocations, and compare them directly with the corresponding standard LoRA baselines? This would make the comparison more convincing.

* How sensitive is the method to the choice of $r_{\min}$? Since $r_{min}$ is an important component in the final allocation formula,
  it seems to directly affect how much freedom remains for importance-driven allocation.

**Limitations:**

yes

**Strengths And Weaknesses:**

## Strengths

* The motivation is natural and the method is simple and direct. Confidence-guided probing, and proportional allocation based on Taylor scores form a clear pipeline that is easy to understand and implement.

* The approach is cheap and practical. The proposed method can probe once and then apply across rank budgets and LoRA backends.

* The experiments are fairly solid. The paper covers multiple models, multiple tasks, and multiple LoRA variants, and the ablation studies are reasonably thorough.

## Weaknesses

* On many tasks, the improvements are relatively minor. The gains are generally consistent, but often modest. For example, on the GLUE benchmark, the performance improvement over many subtasks is minor, and some are even worse than the uniform LoRA.

* The justification for assigning larger rank to modules with larger Taylor saliency is still largely heuristic. The paper gives reasonable intuition and empirical support, but it does not yet fully establish that larger saliency should directly imply larger rank allocation in a more causal sense.

* Relatedly, the paper should report the actual number of trainable parameters selected by the proposed allocation more explicitly when comparing against standard LoRA. The setup claims identical total parameter budgets for the uniform-rank baselines, but the method itself allocates rank across layers of different sizes, so it would be useful to verify more directly that the performance gains are due to better allocation rather than differences in effective trainable parameter count.

---

> ### Author Rebuttal · Authors · 2026-03-30
>
> We appreciate the reviewer's positive assessment of the overall framework, the intuition of the method, and the completeness of the experiments. The main concern appears to be parameter fairness and the interpretation of the allocation mechanism, so we focus this part of the response on these two issues.
>
> ### 1 & 2 On whether “sum Taylor” merely favors larger and more parameter-expensive modules, thereby implicitly increasing effective trainable parameters, and on the actual trainable parameter counts
>
> We believe this is an important and well-motivated concern. The reviewer points out a key fairness issue: when module sizes differ, the same total rank budget does not automatically imply the same trainable parameter count. Therefore, if we only compare rank allocations without explicitly comparing the final trainable parameter counts, it is indeed difficult to rule out the alternative explanation that part of the gain may come from a larger effective number of trainable parameters.
>
> We also agree that the current wording of Sec. 4.3.3, especially the sentence “larger FFN layers consistently receive higher allocation,” is not precise enough and may give the impression that the final allocation mechanically favors larger modules. This is not the conclusion we intend to convey. More precisely, that sentence refers to an observation at the raw saliency-analysis level rather than to the direct rule of the final planner. The final `CSPLoRA` allocation further goes through smoothing and budget-constrained post-processing, so it is not obtained by a simple rule of “the larger the module, the higher the rank.”
>
> To address this concern directly, we conducted an explicit parameter audit. To keep all newly added evidence internally consistent, the new results reported below were all rerun on `RTX 5090` environment. Under the representative setting of `LLaMA-2-7B + commonsense + LoRA backend`, standard uniform `LoRA r=8` has `19.99M` trainable parameters, while the corresponding `CSPLoRA` allocation has `19.98M` trainable parameters. The difference is only `6,912` parameters, i.e., about `-0.03%`, which is too small to account for the downstream performance gap.
>
> At the same time, under this nearly matched-parameter setting, `CSPLoRA` still performs better. The corresponding `5-task` results are shown below. This also directly addresses the reviewer’s request to explicitly report the actual trainable parameter counts and compare them against standard LoRA.
>
> | Method  | Trainable Params | BoolQ |  PIQA |  HeSw |  WiGr | ARC-C |       Avg |
> | ------- | ---------------: | ----: | ----: | ----: | ----: | ----: | --------: |
> | LoRA    |           19.99M | 67.89 | 80.47 | 78.06 | 76.56 | 63.82 |     73.36 |
> | CSPLoRA |           19.98M | 68.53 | 79.92 | 81.83 | 76.95 | 63.31 | **74.11** |
>
> Therefore, under this representative setting, the current evidence does not support the interpretation that the gain mainly comes from assigning rank to larger and more expensive modules and thereby implicitly increasing the effective trainable parameter count. A more appropriate interpretation is that the gain comes from the full planning framework rather than from raw `sum Taylor` alone.
>
> In addition, to make the final `r` allocation more concrete, we further report several modules with the highest and lowest ranks in this representative final allocation. This example shows that the final allocation is not a direct mapping from module size, but the result of the complete planning pipeline. We will also revise the wording in Sec. 4.3.3 accordingly to make this point clearer in the paper.
>
> | Group  | Layer | Module |    r |
> | ------ | ----: | ------ | ---: |
> | Top    |    10 | k      |   11 |
> | Top    |    10 | o      |   11 |
> | Top    |    10 | q      |   11 |
> | Top    |    10 | v      |   11 |
> | Top    |    11 | k      |   11 |
> | Bottom |     0 | gate   |    4 |
> | Bottom |     0 | k      |    4 |
> | Bottom |     0 | up     |    4 |
> | Bottom |     1 | k      |    4 |
> | Bottom |     0 | q      |    3 |
>
> ### 3 On the sensitivity to `r_min`
>
> We also conducted a sensitivity analysis on `r_min`, with the results shown below:
>
> | r_min | BoolQ |  PIQA |  HeSw |  WiGr | ARC-C |       Avg |
> | ----: | ----: | ----: | ----: | ----: | ----: | --------: |
> |     0 | 69.02 | 80.25 | 78.07 | 77.43 | 64.08 |     73.77 |
> |     1 | 68.26 | 80.47 | 80.88 | 76.16 | 63.48 |     73.85 |
> |     2 | 68.53 | 79.92 | 81.83 | 76.95 | 63.31 | **74.11** |
> |     4 | 68.47 | 79.76 | 79.36 | 76.48 | 63.48 |     73.51 |
>
> We add this result in the rebuttal to respond more directly to the reviewer’s question on `r_min`. We will also incorporate this sensitivity analysis into the revised paper and add a clearer explanation of the `r_min` setting.

---

> > ### Author Rebuttal · Reviewer_Wxe6 · 2026-04-04
> >
> > Thank the authors for the rebuttal. I think my main concern on the potential unfairness of the number trainable parameters has been generally resolved. The authors made a clear explanation on this and provided adjusted experiment results. I am glad to raise my score.

---

### Decision · Program_Chairs · 2026-04-30

**Decision:**

Accept (regular)

**Comment:**

The paper introduces CSPLoRA, a framework for pre-training rank allocation in LoRA models using confidence-guided probing. The authors effectively utilized the rebuttal phase to address key concerns.

The authors successfully validated parameter fairness and expanded the empirical scope by adding T5 (encoder-decoder) results, strengthening the paper's claims. However, the core limitation remains: while the method is practical, the performance gains over strong baselines like GoRA are marginal in specific benchmarks, and the necessity of the confidence-weighting component appears less critical than initially claimed.

Given the solid technical validation and the practical value of reusable structural priors, I recommend a Weak Accept.